# LLMCloudHunter: Harnessing LLMs for Automated Extraction of Detection Rules from Cloud-Based CTI

## ABSTRACT

As the number and sophistication of cyber attacks have increased, threat hunting has become a critical aspect of active security, enabling proactive detection and mitigation of threats before they cause significant harm. Open-source cyber threat intelligence (OSCTI) is a valuable resource for threat hunters, however, it often comes in unstructured formats that require further manual analysis. Previous studies aimed at automating OSCTI analysis are limited since (1) they failed to provide actionable outputs, (2) they did not take advantage of images present in OSCTI sources, and (3) they focused on on-premises environments, overlooking the growing importance of cloud environments. To address these gaps, we propose LLMCloudHunter, a novel framework that leverages large language models (LLMs) to automatically generate generic-signature detection rule candidates from textual and visual OSCTI data. We evaluated the quality of the rules generated by the proposed framework using 20 annotated real-world cloud threat reports. The results show that our framework achieved a precision of 83% and recall of 99% for the task of accurately extracting API calls made by the threat actor and a precision of 99% with a recall of 97% for IoCs. Additionally, 99.18% of the generated detection rule candidates were successfully compiled and converted into *Splunk* queries.

## KEYWORDS

Cyber threat intelligence (CTI), Large language model (LLM), Threat hunting, Cloud, Sigma rules

## 1 INTRODUCTION

The rapid evolution of technology, digitization, and application development has been accompanied by an increase in the number of cyberattacks [27], raising concerns about the security risks associated with these advancements. In the face of these concerns, organizations have adopted dynamic defensive strategies in addition to the traditional reactive measures employed [22]. One such strategy is threat hunting, a proactive approach aimed at searching for and mitigating undetected threats in a network or system [16]. Threat hunters try to minimize the damage caused by threat actors by shortening the time window between intrusion and discovery [7]. In their comprehensive survey, Nour et al. [22] stated that the threat hunting methodology consists of three main principles: (1) formulating and testing hypotheses about the threat actor and their actions; (2) utilizing existing information for an intelligence-driven investigation; and (3) leveraging data analysis techniques and machine learning algorithms to effectively handle vast amounts of data.

The second principle involves collecting and analyzing publicly available information about potential and active threats from blogs, forums, and other digital sources. Open-source cyber threat intelligence (OSCTI) is one of the most commonly used sources of information among security personnel according to the SANS 2023 CTI survey [34]. However, various challenges arise when using OSCTI. The first and main challenge is that OSCTI often comes in non-uniform and unstructured formats, such as text and images, rather than more actionable information/data (e.g., detection rules) [31]. As a result, manual analysis by human experts is required to derive meaningful and actionable insights [30]. Another challenge is the increasing amount of available information (i.e., CTIs), necessitating the automation of OSCTI analysis [27].

Previous studies on threat hunting introduced various methodologies, some of which incorporated natural language processing (NLP) techniques, to automate the extraction and enrichment of information from OSCTI textual data. However, the methods presented in these studies suffer from three main limitations: (1) they provide structured but limited insights, such as identified entities and their relationships or attack techniques, necessitating further processing to generate actionable outputs; an exception is the approach presented by Gao et al. [13], in which the authors developed proprietary, non-standard graph-based queries using static rules (regexes) that require substantial customization for application with standard tools and on-premises environments; (2) these studies, including the work of Gao et al., do not take advantage of visual components, such as images, which may be present in OSCTI data; and (3) many of these methodologies were primarily developed for on-premise environments, limiting their effectiveness and relevance in cloud-centric environments.

Cloud computing has become an integral component in the modern enterprise landscape, valued for its scalability, cost-effectiveness, and flexibility [32]. It employs a shared responsibility model for security, in which both the provider and the consumer play roles in securing cloud infrastructure and cloud-delivered applications [3]. This model presents unique challenges in threat hunting, as traditional security methodologies often fall short in addressing the dynamic and distributed nature of cloud environments [41]. Among these challenges is the fact that in some cloud technologies (e.g., serverless), access to data for threat hunting is limited to application-level logs (APIs, storage access, etc.), and important infrastructure-(system)-level data (e.g., virtual machines and network) can only be accessed by the cloud provider [42]. This is exacerbated by the fact that the exploitation of cloud-based threat intelligence has not yet reached maturity. The work of Fengrui and Du [11] is the only study that extends beyond on-premise OSCTI, however rather than providing actionable output, their framework extracts MITRE ATT&CK tactics, techniques, and procedures (TTPs) [1]. These gaps highlight the need for innovative OSCTI analysis approaches suited to the unique security challenges of cloud environments; such challenges can be addressed by integrating OSCTI analysis results within practical, actionable security measures [17].

In this paper, we present LLMCloudHunter, a novel framework that leverages pretrained large language models (LLMs) to generate detection rule candidates from unstructured OSCTIs automatically. LLMCloudHunter generates *Sigma* rule [38] candidates from both textual and visual cyber threat information, using an innovative,

automated data extraction and processing framework that leverages LLMs and employs various techniques to address their limitations (e.g., unstructured output and hallucinations).

*Sigma* rules, provided in a generic and open signature format written in YAML, enable the creation and sharing of detection methods across security information and event management (SIEM) systems. Fig. 1 presents our LLM pipeline for *Sigma* candidate generation; as can be seen, textual and visual OSCTI data is processed first, converting it into semi-structured paragraphs in the preprocessing phase. It then extracts API calls (that are unique entities to threat hunting in cloud environments) and MITRE ATT&CK TTPs from the paragraphs and generates initial *Sigma* candidates (in the Paragraph-Level phase). Finally, it consolidates the candidates from all paragraphs, verifies their syntactic and logical correctness, eliminates duplication, and enriches them with identified indicators of compromise (IoCs) (in the OSCTI-Level phase). An example of a *Sigma* rule generated by LLMCloudHunter is illustrated in Listing 1, with a demonstration of its generation process in Appendix C.

We evaluated the efficacy and precision of the *Sigma* candidates generated using 20 cloud-related OSCTI sources that we identified. The evaluation was performed using common entity and relationship extraction metrics, and the results were validated against a ground truth carefully defined by our research team. Additionally, we introduced a set of criteria specifically designed to test each *Sigma* candidate's functionality in the operational context of OSCTI. This evaluation ensures that the rules generated not only meet syntactic standards but are also operationally effective in addressing the dynamic and complex nature of cloud-based cyber threats. We also conducted an ablation study, systematically removing components of the framework to pinpoint their individual contributions to LLMCloudHunter's overall efficacy. The results show that our framework achieved a precision of 83% and recall of 99% for the task of accurately extracting threat actors' API calls, and a precision of 97% with a recall of 97% for IoCs. Moreover, 99.18% of the generated *Sigma* candidates were successfully converted into *Splunk* queries. In terms of overall performance, i.e., including the extraction of API calls, IoCs, MITRE ATT&CK TTPs, and request parameters, our framework achieved 85% and 88% precision and recall, respectively.

To summarize, the main contributions of this paper are: (1) *A novel LLM-based framework for the automatic generation of Sigma candidates from unstructured OSCTI*, which integrates both textual and visual information. While our framework focuses on cloud environments, it can be adapted for use with on-premise-related CTI. LLMCloudHunter utilizes a pretrained LLM, thus providing flexibility in updating the underlying LLM, and does not require "heavy" model training. (2) *An annotated dataset* (used for the evaluation of our framework) consisting of 20 cloud-related OSCTI posts, complete with entities and their relationships, as well as *Sigma* rules. (3) *Insights on the application of LLMs for complex NLP tasks in the field of cybersecurity*, pertaining to prompt engineering techniques and the effective use of models' features and parameters. (4) *A comprehensive evaluation* that assesses the accuracy and correctness of the *Sigma* candidates generated. (5) *We make both our code and cloud CTI dataset available to the research community* on GitHub.[1]

---

[1]To preserve anonymity, the code and dataset will be available upon paper acceptance.

```
title: Access to Terraform File from Malicious IPs
description: Detects requests for terraform.tfstate file
  from known malicious IPs. This file contains sensitive
  infrastructure information and secrets, indicating
  potential compromise or unauthorized access.
references:
    - https://sysdig.com/blog/cloud-breach-terraform-data-
    theft/
    - https://docs.aws.amazon.com/AmazonS3/latest/API/
    API_GetObject.html
author: LLMCloudHunter
tags:
    - attack.collection
    - attack.t1530
logsource:
    product: aws
    service: cloudtrail
detection:
    selection_event:
        eventSource: s3.amazonaws.com
        eventName: GetObject
        requestParameters.key: terraform.tfstate
    selection_ip_address:
        sourceIPAddress:
            - 80.239.140.66
            - 45.9.148.221
            - 45.9.148.121
            - 45.9.249.58
    condition: selection_event and selection_ip_address
falsepositives:
    - Automated CI/CD pipeline operations
    - DevOps engineers manually running Terraform commands
level: high
```

**Listing 1: A Sigma rule generated by LLMCloudHunter.**

## 2 RELATED WORK

In this section, we provide a brief overview of recent studies focused on analyzing unstructured OSCTI analysis. A detailed description of related work is provided in Appendix A.

Earlier works have extensively utilized NLP techniques for OSCTI analysis [4, 28, 35–37]. These methods leveraged advanced NLP models to extract actionable insights from OSCTI text. However, to adapt these models to the cyber threat domain, a significant amount of preprocessing and fine-tuning is required. While the approach implemented by TTPDrill [15] and THREATRAPTOR [13] reduces the need for extensive model training, it is not flexible, and significant customization is needed for use in cloud environments. This is due to fundamental differences in terminology and data types between traditional on-premise environments and cloud environments, as well as the dynamic nature of cloud architectures, which continuously evolve with new services and configurations.

The introduction of LLMs has led to a paradigm shift in OSCTI processing, with research demonstrating their ability to extract meaningful and structured data from OSCTI text. Utilizing GPT-3.5, Purba and Chu [29] and Siracusano et al. [39] addressed tasks ranging from the extraction of IoCs to the generation of structured CTI format (e.g., STIX), respectively, while Liu and Zhan [20] applied ChatGPT to construct graphical representations of OSCTI data. Hu et al. [14] and Fengrui and Du [11] expanded upon these capabilities by utilizing both pretrained and fine-tuned LLM models. They employed GPT-3.5 and ChatGPT for data annotation and augmentation, respectively, to prepare datasets for fine-tuning the LLaMA2-7B model. Hu et al. [14] applied the fine-tuned LLaMA2-7B to construct

knowledge graphs, while Fengrui and Du [11] focused on TTP classification. In this research, we are the first to develop an end-to-end framework based on a pretrained LLM, demonstrating the potential of LLMs in processing OSCTI and generating actionable *Sigma* rules. Moreover, our framework integrates visual analysis capabilities, expanding the scope of OSCTI analysis beyond previous text-centric methodologies. By leveraging pretrained LLMs, we avoid the need for rule-based methods or training customized models with dedicated datasets. Our framework also focuses on generating rules for cloud environments, which has not been addressed before.

In terms of OSCTI datasets, in contrast to prior studies that used semi-structured and on-premise-related datasets, we use 20 unstructured, publicly available *cloud-based* posts and reports sourced from various publishers. These OSCTI reports, which describe AWS cloud incidents, were systematically annotated by our research team to develop a robust ground truth for development and evaluation.

Previous studies produced a variety of outputs with different levels of utility and applicability. This includes extracting IoCs [19, 29], TTPs [11], and structured representations using the STIX format [12, 15]. More advanced approaches were used to create threat behaviour graphs [13, 37] and knowledge graphs [4, 6, 14, 20, 28, 35, 37]. While the approaches highlighted above provide valuable contextual information, further processing is required to transform the representations into actionable defense mechanisms. To address this, in their study, Gao et al. presented a framework for converting OSCTI data into a threat behavior graph and associated domain-specific queries. The detection rule candidates generated by LLMCloudHunter, however, are in the known open-source *Sigma* structure. This widely used generic signature format is inherently suitable for integration in various application environments and SIEMs. By capturing the entities, relations, IoCs, and TTPs identified in OSCTI, LLMCloudHunter translates threat intelligence into applicative *Sigma* candidates.

## 3 PROPOSED METHOD

In this section, we present our proposed framework, LLMCloud-Hunter, and how it leverages OpenAI's GPT-4o [25] model to process cloud-based OSCTIs and generate *Sigma* candidates. LLM-CloudHunter's pipeline (see Fig. 1) consists of three main phases: *Preprocessing*, *Paragraph-Level Processing*, and *OSCTI-Level Processing*; these phases are described in the subsections that follow.

**Relevant Entities for Threat Hunting in Cloud Environments.** The atomic units in cloud application logs are cloud API calls, which describe system and application activities that potentially provide traces of threat behavior. An example of an API call may be the *GetFunction* action, which requests information about a function. Therefore, the information used to generate *Sigma* candidates for threat hunting in cloud environments includes entities such as IP addresses and user agents, similar to on-premise environments, as well as API calls that are unique to cloud environments.

We differentiate between primary (essential) entities and contextual entities. Primary entities are required for the correct execution of generated *Sigma* candidates in SIEM systems. A mistake in extracting a relationship that includes a primary entity will result in incorrect "hunting" activity. Primary entities in cloud environments include API calls (e.g., *GetFunction*) as well as the request parameters of that API call (e.g., *requestParameters.functionName: respondUser*),

IoCs (including IP addresses and user agents), log source (e.g., *AWS CloudTrail*), and event source (e.g., *lambda.amazonaws.com*). Contextual entities do not impact the correctness of the detection rule logic; however, they provide additional contextual information to the threat hunter, making the investigation of a case more efficient. Contextual entities include the title and description of the *Sigma* rule, TTPs, false positives, and criticality level.

### 3.1 OSCTI Preprocessing

OSCTI varies in terms of the type and format, depending on the publishing platform, the author, the nature of the collected information, and its intended purpose. Due to this lack of uniformity, preliminary steps must be performed to standardize the format. Such steps enable the data to be automatically and effectively handled by subsequent processing components. The preprocessing converts the HTML content into a structured markdown format, which has been shown to improve LLM task performance [18]. Additionally, our framework uniquely handles image extraction, classification, and transcription—a novel approach compared to related works.

**Downloader and Parser.** The automated OSCTI preprocessing phase begins by downloading and parsing the OSCTI HTML code (A in Fig. 1), using web scraping and processing tools such as Selenium [21] and BeautifulSoup [33], followed by additional reformatting techniques (e.g., regex) to ensure a valid OSCTI markdown output. By examining the web page elements, LLMCloudHunter pinpoints the beginning and end of the relevant content, excluding irrelevant elements (such as sidebars and advertisements). In the next step, these HTML layout elements are converted into a unified markdown based on the following guidelines: (1) Preserve spacing to separate content types such as paragraphs and code sections, maintaining their original layout. (2) Mark headings (h1, h2, etc.) to maintain the hierarchical structure of the original HTML content. (3) Parse HTML code encompassing tables and nested lists to preserve their structural properties. For example, a tab character is employed in lists to signify nested items, whereas in tables, the '|' symbol is used to demarcate columns. (4) Identify and embed image URLs as placeholders within the text, positioning them according to their original placement in the report.

After converting the HTML into a markdown, we employed a targeted approach to exclude non-essential content (including headings, subheadings, and the corresponding paragraphs). Such content is identified by indicative keywords that suggest repetitive and redundant information. Examples of this type of content include overviews, recommendations, and concluding paragraphs. For instance, if a 'recommendations' paragraph appears under an h2 heading, we remove the paragraph and any subsequent content until the next h2 (or h1) heading is encountered, as recommendations are not part of the attack description and often include marketing content. This approach effectively removes non-essential or duplicated content nested under the identified headings. The filtered version of the output is then passed on to the next component in the framework. The full output, which includes all content, will be used in the *OSCTI-Level Processing* phase.

**Image Analysys.** Continuing with the *Preprocessing* phase, each image is first classified by the *Image Classifier*(B in Fig. 1) using a *classification prompt* as either an informative image (e.g., screenshots, charts, diagrams, and tables containing information related

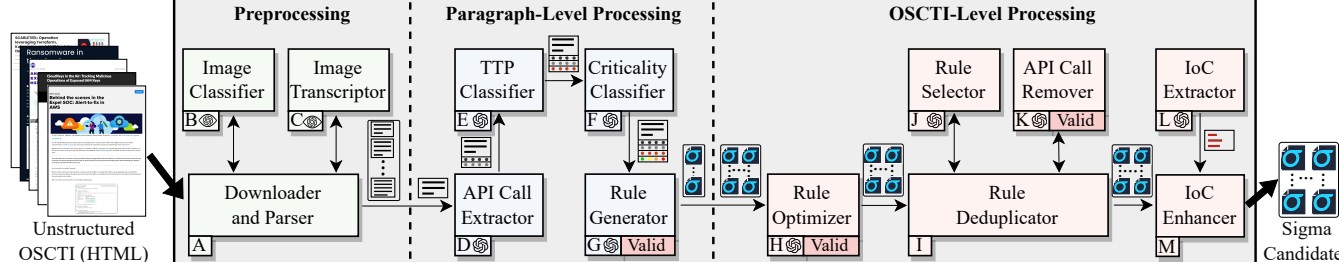

**Figure 1: Overview of the LLMCloudHunter framework.**

to the OSCTI content) or non-informative one(e.g., decorative art, advertisements, logos, or generic symbols). The prompt includes the text of the paragraph in which the image is located in the OSCTI as context to assist the LLM in determining its classification. Along with the classification, we requested the LLM to explain the image classification to facilitate human validation during testing. If an image is classified as informative, it is then passed to *Image Transcriptor* (C in Fig. 1). It is processed using a *transcription prompt* to extract and convert its content into the most appropriate markdown format (e.g., lists and code). The extracted text is integrated into the OSCTI formatted text in its original location, preserving the report's context/flow and enhancing it with critical details, such as API calls and IoCs. By adopting this comprehensive image processing approach, the framework increases the accuracy of extracted information and introduces a novel method in OSCTI analysis (See ablation study 4). Unlike previous works, which have overlooked the potential value of visual data, our framework integrates relevant images into the analytical pipeline, ensuring that no critical information is missed. The image classification and transcription prompts are provided in Appendix F.

## 3.2 Paragraph-Level Processing

After preprocessing the OSCTI, the next phase in the LLMCloud-Hunter framework is *Paragraph-Level Processing*. In this phase, LLMCloudHunter first identifies key entities: API calls, MITRE ATT&CK TTPs, and threat event criticality levels. These entities are then used to enrich the formatted paragraphs, from which LLM-CloudHunter generates initial *Sigma* candidates. To perform these complex tasks, LLMs require carefully defined steps of accurate information extraction and effective data linkage. Our experiments showed that segmenting the OSCTI text into manageable chunks (i.e., paragraphs) enhances the efficiency of the tasks involved in *Sigma* candidate generation. This approach aligns with the natural structure of writing, organizing information into semantically distinct paragraphs, which narrows the model's focus and minimizes errors. Additionally, we leverage parallelization by processing these paragraphs concurrently to boost processing speed significantly.

**API Call Extractor.** The *Paragraph-Level Processing* phase starts with the *API Call Extractor* (D in Fig. 1), which analyzes paragraphs from the OSCTI formatted text that were generated in the previous phase and extracts both explicitly mentioned and implicitly referred API calls in each paragraph (this process is depicted in the flowchart presented in Appendix F). To improve the model's output reliability, mitigate hallucinations (e.g., referencing nonexistent events), and

prevent the omission of API calls, we incorporate a majority voting mechanism to ensure higher accuracy and confidence in identifying and extracting relevant API calls.

The operational flow begins with the *explicit API call extractor*, where a dedicated prompt instructs the LLM to extract all explicitly mentioned API calls in the paragraph. This operation is executed $N_{explicit}$ times, with API calls that exceed the $T_{explicit}$ threshold selected for subsequent analysis. Only paragraphs containing API calls that meet the $T_{explicit}$ are kept; the rest are discarded.

Then, paragraphs that are found to contain explicit API calls undergo more nuanced extraction by the *Implicit API Call Extractor*. In this step, we utilized the LLM to perform a deeper analysis to infer API calls suggested indirectly by the OSCTI author. For example, operational descriptions such as performing a *sync* action on an S3 bucket should be mapped to the *ListBuckets* and *GetObject* API calls. Due to the complexity of identifying these implicit API calls, this step is executed $N_{implicit}$ times, where $N_{implicit}$ is set to twice the number of $N_{explicit}$ iterations performed. Similar to the explicit API call extraction process, paragraphs are analyzed for implicit API calls that meet the $T_{implicit}$ threshold. However, paragraphs without any implicit API calls are not discarded, as they still have some value due to their explicit API call content.

**TTP Extractor.** This component (E in Fig. 1) analyzes the extracted API calls, mapping them to cloud-based MITRE ATT&CK tactics, techniques, and sub-techniques. It utilizes a detailed prompt, which includes mapping cloud tactics to techniques and techniques to sub-techniques (in JSON format), along with illustrative examples of effective and ineffective mappings. This integrated approach not only enhances the accuracy of TTP assignments but also safeguards against model hallucinations. Each API call is evaluated in its specific context to assign the most precise and relevant TTPs. While these TTPs do not directly alter the detection logic of the *Sigma* candidates, they play a critical role in understanding the structure of the attack and classifying its various stages.

**Criticality Classifier.** This component (F in Fig. 1) estimates the severity of each *Sigma* candidate. It uses a single prompt, which includes the paragraph markdown along with the extracted API calls and TTPs, to classify API calls into appropriate criticality levels based on their context. The prompt guides the LLM by providing examples (zero-shot learning), helping emphasize each API call's potential impact, malicious use, and monitoring importance.

**Rule Generator.** The last component in the *Paragraph-Level Processing* phase (G in Fig. 1) receives as input a list of identified API calls, their criticality, and corresponding TTP assignments, bundled

with the paragraph markdown. The LLM processes this enriched input using the *Rule Generator* prompt (the full prompt is provided in Appendix F). This prompt defines the LLM's role as a cybersecurity analysis tool that specializes in generating *Sigma* rules from OSCTI text. This approach aims to leverage extracted AWS API calls to enrich paragraphs and transform them into *Sigma* candidates. This, in turn, enables the detection of similar activities or patterns in log files. The generation prompt includes several important instructions:

- Each API call provided (along with its TTPs) must be included in the *Sigma* candidates, but not more than once, to avoid the omission of important details and duplications.
- Paying attention to small details is extremely important as they can improve the detection specificity of the *Sigma* candidates.
- *Sigma* candidates with the same attack patterns and stages (i.e., their TTPs) should be merged and vice versa.
- *Sigma* candidates must align with the specific terminology and functionality of AWS environments to ensure relevance.
- The output (i.e., LLM response) is required to be in a uniform and interpretable format. We used JSON format since it is a built-in feature available through the OpenAI API [24].

**Rule Validator.** Once *Sigma* candidates are generated, a validation function is applied to ensure that the output complies with the *Sigma* standard structure (YAML). This function is denoted as *Valid* in Fig. 1, and is executed by each component that produces rules using LLM. This validation process involves sanitizing too specific or extraneous fields, such as *errorcode*, *errormessage*, and explicit resource names, to enhance the applicability of the rules. It also reformats the syntax to ensure the validity of *<key:value>* pairs and verifies metadata, including author names, reference URLs, and dates. This function safeguards the integrity and consistency of the *Sigma* candidates by eliminating redundant attributes and correcting structural flaws.

## 3.3 OSCTI-Level Processing

The final phase in the LLMCloudHunter framework aggregates *Sigma* candidates generated from individual paragraphs to produce a consolidated and optimized set of detection rules, enabling holistic processing and enrichment. It takes the collected *Sigma* candidates from all processed paragraphs and outputs a final, optimized set free of redundancies and enriched with IoCs.

**Rule Optimizer.** The first component (H in Fig. 1) in the *OSCTI-Level Processing* phase is designed to improve *Sigma* candidates' detection logic. In this component, the LLM processes the validated *Sigma* candidates concurrently to enhance the speed and efficiency of the optimization process. A designated prompt, along with optimization examples, guides the LLM to ensure that the detection criteria are clear and aligned with their intended purpose. The optimization process includes the following tasks:

- **Unification** - merges *selection* fields that match identical detection criteria, i.e., those sharing the same filtering logic. For example, consider the *Sigma* rule in Listing 1, which detects access to a certain file from malicious IP addresses. Assume this *Sigma* rule includes another *selection* field with the same event source, event name, and request parameter (*s3.amazonaws.com*, *GetObject*, and *terraform.tfstate*, respectively) but adds an additional request parameter: *requestParameters.bucket: Starak*. When

performing the unification task, the *Rule Optimizer* combines these two *selection* fields into a single *selection* that encompasses all relevant fields: *eventSource*, *eventName*, *requestParameters.key*, and *requestParameters.bucket*. This unification ensures that the rules are comprehensive and free of redundancy by merging overlapping criteria while preserving their original integrity.

- **Separation** - Splits disjoint *selection* fields that share some detection criteria but have misaligned logic. For example, consider the *Sigma* rule in Listing 1. Assume that the initial *Sigma* rule incorrectly included two additional unrelated fields: *eventSource: iam.amazonaws.com* and *eventName: PutUserPolicy* in the same existing *selection* field. The *Rule Optimizer* would recognize that these fields are unrelated to the original detection logic and would separate them into a new *selection* field. Then, it would update the *condition* field to search for either the first *selection* or the newly created second *selection*. This separation ensures the rule remains accurate and logically consistent by distinguishing between different detection criteria.

---

**Algorithm 1** Rule Deduplicator.

---

**Input:** A set of *Sigma* candidates *osctiRules*
**Output:** Modified *osctiRules*

1: *osctiAPIs* ← ExtractAPIs(*osctiRules*)
2: **for** each *osctiAPI* ∈ *osctiAPIs* **do**
3:     *commonRules* ← GetCommonRules(*osctiRules*, *osctiAPI*)
4:     *selectedRule* ← RuleSelector(*commonRules*, *osctiAPI*)
5:     *rulesToAdjust* ← *commonRules* − *selectedRule*
6:     **for** each *ruleToAdjust* ∈ *rulesToAdjust* **do**
7:         *ruleAPIs* ← ExtractAPIs(*ruleToAdjust*)
8:         **if** |*ruleAPIs*| = 1 **then**
9:             *osctiRules* ← *osctiRules* − *ruleToAdjust*
10:        **else**
11:            APICallRemover(*ruleToAdjust*, *osctiAPI*)
12:        **end if**
13:    **end for**
14: **end for**

---

**Rule Selector.** This component (J in Fig. 1) refines the *Sigma* candidate set by selecting the most suitable rule among those containing the same API call. It uses prompts to evaluate the specificity and context of each rule, prioritizing those with detailed criteria directly linked to the API call. If multiple rules are equally specific, the context (the paragraph of which they have been generated) is used to make the final selection.

**API Call Remover.** Following the *Rule Selector*'s identification of the best rule, the *API Call Remover* (K in Fig. 1) edits other rules containing the same API call. It systematically preserves each rule's structure while removing the redundant API call. If a rule solely depends on the API call being removed, it is discarded entirely.

**Rule Deduplicator.** Working with the *Rule Selector* and *API Call Remover*, the *Rule Deduplicator* (I in Fig. 1) finalizes the *Sigma* candidate set by eliminating overlaps as the depicted in Algorithm 1. It maps event names to rule indices and retains only the most comprehensive rule for each detection scenario, resulting in a precise and non-overlapping set of *Sigma* candidates.

**IoC Extractor.** This component (L in Fig. 1) parses OSCTI texts to identify and extract explicit IoCs, notably IP addresses and user

agents pertinent to AWS CloudTrail logs. Its input is the full markdown of the OSCTI created by the *Downloader and Parser*, along with an instruction prompt. This prompt guides the LLM to focus on paragraphs typically containing IoCs (e.g., conclusion, findings, or IoC sections). Additionally, the LLM is instructed to ensure that all IoCs are identified and to convert obfuscated IP addresses and user agents to standardized formats.

**IoC Enhancer.** Following the extraction of IoCs by the *IoC Extractor*, this component (M in Fig. 1) integrates the extracted IoCs into all *Sigma* candidates, enhancing their detection capabilities while maintaining flexibility for analysts. The IoCs (IP addresses and user agents) associated with the threat actor are added to each *Sigma* candidate as optional detection filters. The *IoC Enhancer* introduces new *selection* fields for each type of IoC. For instance, when an IP address is extracted (*198.51.100.1*), the *selection_ioc_ip* field is added: *selection_ioc_ip: sourceIPAddress: 198.51.100.1*. Similarly, when a user agent is extracted (*Mozilla/5.0*), the *selection_ioc_ua* field is introduced: *selection_ioc_ua: userAgent|contains: Mozilla/5.0.* The *|contains* operator is used to improve string matching flexibility, allowing for variations (e.g., different versions). After adding these IoC selections, the *IoC Enhancer* updates the *condition* field of each *Sigma* candidate to include the IoCs as optional criteria. If the original condition was: *condition: selection*, it is modified to: *selection and (selection_ioc_ip or selection_ioc_ua)*. This ensures that an event must meet the original detection criteria (e.g., specific API calls and event sources) and either the IP address or user agent IoC. By integrating IoCs in this way, the rules become more accurate in detecting activities associated with the threat actor. Importantly, since the IoCs are added as optional filters, analysts can easily adjust the rules to suit their needs. If the IoCs lead to false positives or become irrelevant, analysts can remove or modify these conditions without altering the core detection logic. This approach maintains transparency of information passed from OSCTI to the *Sigma* rules while ensuring the *Sigma* candidates remain adaptable for various use cases.

## 4 EVALUATION

In this section, we describe the creation of an annotated benchmark dataset and present the methodology and metrics used to evaluate the efficacy and accuracy of the Sigma candidates generated by LLMCloudHunter. We present the results of our evaluation, which also includes an ablation study in which we analyze the impact of each of the framework's components on the overall performance.

### 4.1 Dataset

We collected 20 cloud environment OSCTIs published by different vendors. Table 6 in Appendix 6 provides a description of the OSCTIs, including the number of images, token sizes, number of API calls, and their technical complexity. To establish the dataset's ground truth, a team of threat hunting and cloud security experts thoroughly analyzed each OSCTI's content. The team (1) identified and extracted the entities described in the OSCTI and (2) identified the relevant inter-entity relationships essential for creating coherent and meaningful *Sigma* candidates. The list of extracted entities and inter-entity relationships is provided in Table 1. We categorized the entities and relationships into two main groups:

**(1) Detection:** These are essential elements required to form a correct *Sigma* rule for detecting threat actor actions. This category includes detection entities and their associated *Detection Field Name* relationships, which are crucial for measuring *key:value* placements in rule generation.

**(2) Informative (MITRE ATT&CK Tags):** Entities not directly involved in detection logic but relevant for adding context to the alerts raised by the rules, based on associated TTPs.

| Entity | Relationship |
|---|---|
| **Detection Entities and Relationships** | |
| API Call | Detection Field Name ↔ Detection Entity |
| Log Source | API Call ↔ Log Source |
| API Source | API Call ↔ API Source |
| IoC | API Call ↔ IoC |
| Other | API Call ↔ Other |
| **MITRE ATT&CK Entities and Relationships** | |
| Technique | API Call ↔ Technique |
| Sub-Technique | API Call ↔ Sub-Technique |

**Table 1: Entity types and relationships.**

### 4.2 Evaluation Metrics

We evaluated our framework's performance using a comprehensive set of metrics designed to assess both the extraction of entities and relationships from OSCTIs and the functionality of the generated *Sigma* candidates.

**Entity and Relationship Extraction Metrics:** We utilized common entity and relationship extraction metrics, as done in prior studies [4, 6, 11–13, 20, 28, 29, 35–37, 39], to assess our framework's performance, validating the results against the ground truth defined by our research team. The metrics used to assess LLMCloudHunter's performance in extracting and identifying the entities and inter-entity relationships in the OSCTI are the precision (P), recall (R), and F1 score (F1) weighted by the total number of entities/relationships of each type, denoted as '#' (since each OSCTI has a different number of entities/relationships). By calculating these metrics separately for each entity and relationship type, we can pinpoint areas of strength and identify opportunities for improvement.

To evaluate the functionality, logical validity, and relevance of the *Sigma* candidates generated by LLMCloudHunter, we defined the following criteria. These metrics were calculated by our research team for each *Sigma* candidate generated:

- **Syntax Correctness** - Assesses whether the generated *Sigma* candidates are syntactically correct and properly formatted, ensuring that a given rule is operational in a SIEM system. We used *Sigma* CLI [2] for compilation and conversion into query languages (e.g., Splunk).
- **Detection Condition Accuracy** - Focuses on the correctness of the *condition* fields, which specify the relationship between various *selection* fields.
- **Criticality Accuracy** - Measures the accuracy of the *level* field of each *Sigma* candidate, which represents the level of importance and urgency of the rule.
- **Descriptive Metadata Alignment** - Evaluates whether the *title*, *description*, and *falsepositives* fields accurately reflect the rule's intended purpose and context.

## 4.3 Results

The results (averaged over all evaluated OSCTIs) of our entity and relationship extraction evaluation are presented in Tables 2 and 3, respectively (detailed results are provided in Appendix B).

**Detection.** We consider API calls and IoCs to be the most important entities for generating practical and relevant *Sigma* candidates. For these two entity types, LLMCloudHunter achieved a weighted precision of 83% with a recall of 99% for the API calls and a precision of 99% with a recall of 97% for the IoCs. In the 'Other' entity category, which includes various entities (e.g., request parameters and IP address), resulted in precision and recall values of 75% and 61%, respectively. The relationship extraction results, which represent LLMCloudHunter's ability to interrelate detection entities to the appropriate fields in *Sigma* rules, achieved an F1 score of 96% for the *Detection Field Name ↔ Detection Entity* relationship.

**Informative (MITRE Tags).** For the extraction of MITRE ATT&CK TTPs, which is known to be a challenging task [8], LLMCloud-Hunter achieved an F1 score of 74% for technique and 81% for sub-technique. Since each technique and sub-technique directly maps to one or more known tactics, this entity becomes redundant. For instance, *'Cloud Service Discovery (T1526)'* maps to the *'Discovery'* tactic, illustrating how tactics can be directly inferred from techniques, rendering the explicit identification of tactics redundant. These results are notable compared to similar works; for instance, Daniel et al. [10] reported a highest F1 score of 0.49 in MITRE tags extraction. The relationship identification results, which represent LLMCloudHunter's ability to interrelate the detection entities to the relevant key in the *Sigma* candidates (*Detection Field Name ↔ Detection Entity*), achieved an F1-score of 96%. Regarding the extraction of MITRE ATT&CK TTPs, LLMCloudHunter achieved an F1 score of 74% for Techniques and 81% for Sub-Techniques, with notably high recall rates of 82% and 90%, respectively. The precision was impacted due to LLMCloudHunter generating more *Sigma* candidates than the ground truth, leading to the creation of additional, more specific tags. This increase in the number of tags stems from LLMCloudHunter's strategy to extract all the threat actor actions, resulting in a higher number of API Calls and, thus, a higher number of false positives when compared to the ground truth, thus lowering the precision. Similarly, in the relationship extraction task, the low precision for MITRE-related relationships can be attributed to the model associating more specific Techniques and Sub-Techniques with the API Calls, which were not always present in the ground truth. While this affects the precision metric, the high recall indicates that LLMCloudHunter successfully captures the relevant TTPs, providing valuable context for threat detection.

In summary, LLMCloudHunter demonstrates strong performance in extracting and identifying key entities and their relationships within OSCTI. While the framework was shown to excel in handling API calls, IoCs, and request parameters, achieving high precision and recall for this, it faces challenges with MITRE ATT&CK TTPs, which impacts the overall performance but does not affect the detection capabilities of the *Sigma* candidates generated.

The results of our *Sigma* candidate evaluation are presented in Table 4. Out of 260 generated candidates, an impressive 99.18% were syntactically correct and operational, showcasing high syntax correctness. The detection condition accuracy was equally noteworthy, with all but one candidate correctly specifying the logical relationships between selection fields, resulting in an accuracy rate exceeding 99%. While the criticality accuracy varied between 75% and 100% across different OSCTIs—with an average of approximately 88% — this suggests that LLMCloudHunter generally assigns appropriate importance levels, though there is room for improvement in aligning more closely with expert assessments. Lastly, the descriptive metadata alignment was exceptional, with most OSCTIs scoring above 95%, demonstrating that LLMCloudHunter effectively generates titles, descriptions, and false positive information that accurately reflect each rule's intended purpose and context.

| | Entity | # | P | R | F1 |
|---|---|---|---|---|---|
| **Detection** | Field Name | 8.20 | 0.85 | 0.85 | 0.85 |
| | API Call | 18.75 | 0.83 | **0.99** | **0.90** |
| | IoC | 9.50 | 0.99 | 0.97 | 0.98 |
| | Log Source | 2.00 | 1.00 | 1.00 | 1.00 |
| | Other | 3.45 | 0.75 | 0.61 | 0.67 |
| **MITRE ATT&CK** | Technique | 6.25 | 0.67 | **0.82** | 0.74 |
| | Sub-Technique | 3.00 | 0.73 | **0.90** | 0.81 |

**Table 2: Entity extraction results.**

| | Relationship | # | P | R | F1 |
|---|---|---|---|---|---|
| **Detection** | Field Name ↔ Detection Entity | 33.00 | 1.00 | 0.93 | **0.96** |
| | API Call ↔ API Source | 17.60 | 1.00 | 0.82 | **0.90** |
| | API Call ↔ IoC | 31.20 | 1.00 | 0.99 | **0.99** |
| | API Call ↔ Other | 5.90 | 0.92 | 0.55 | 0.69 |
| | API Call ↔ Log Source | 31.20 | 1.00 | 0.99 | 0.99 |
| **MITRE ATT&CK** | API Call ↔ Technique | 16.85 | 0.61 | 0.47 | 0.53 |
| | API Call ↔ Sub-technique | 5.15 | **0.92** | 0.69 | 0.79 |

**Table 3: Relationship extraction results.**

| OSCTI ID | #Rules | Executability | Condition Field Accuracy | Criticality Accuracy | Descriptive Metadata Alignment |
|---|---|---|---|---|---|
| 1 | 10 | 9 (90%) | 9 (90%) | 87.50% | 93.75% |
| 2 | 15 | 15 (100%) | 15 (100%) | 90.00% | 95.00% |
| 3 | 15 | 15 (100%) | 15 (100%) | 83.33% | 90.00% |
| 4 | 9 | 9 (100%) | 9 (100%) | 83.33% | 100.00% |
| 5 | 18 | 18 (100%) | 18 (100%) | 86.11% | 100.00% |
| 6 | 14 | 14 (100%) | 14 (100%) | 92.86% | 100.00% |
| 7 | 7 | 7 (100%) | 7 (100%) | 85.71% | 100.00% |
| 8 | 9 | 9 (100%) | 9 (100%) | 83.33% | 100.00% |
| 9 | 4 | 4 (100%) | 4 (100%) | 75.00% | 87.50% |
| 10 | 15 | 15 (100%) | 15 (100%) | 96.43% | 100.00% |
| 11 | 14 | 14 (100%) | 14 (100%) | 82.14% | 96.43% |
| 12 | 18 | 13 (100%) | 13 (100%) | 93.75% | 100.00% |
| 13 | 24 | 24 (100%) | 24 (100%) | 97.92% | 96.88% |
| 14 | 6 | 6 (100%) | 6 (100%) | 83.33% | 100.00% |
| 15 | 4 | 4 (100%) | 4 (100%) | 87.50% | 100.00% |
| 16 | 39 | 39 (100%) | 39 (100%) | 90.38% | 98.08% |
| 17 | 6 | 6 (100%) | 6 (100%) | 90.00% | 100.00% |
| 18 | 6 | 6 (100%) | 6 (100%) | 83.33% | 100.00% |
| 19 | 12 | 12 (100%) | 12 (100%) | 91.67% | 95.83% |
| 20 | 15 | 15 (100%) | 15 (100%) | 100.00% | 96.67% |
| **Weighted Avg.** | 15 | 99.18% | 100.00% | 88.18% | 97.50% |

**Table 4: Sigma candidate evaluation results.**

**Ablation Study Results.** We conducted an ablation study to better understand the impact of LLMCloudHunter's components on its performance. We created three variations of LLMCloudHunter by systematically removing key components and evaluating the performance of each variant. Table 11 in Appendix D summarizes

the different configurations used in the ablation study. The Blind-Hunter variation evaluates the impact of the image processing by *Image Classifier* and *Image Transcriptor*. The NoAPIHunter variation is designed to evaluate the impact of the *API Call Extractor* and *TTP Classifier* components (D and F in Fig. 1, respectively); ;the UnoptimizedHunter variation aims to evaluate the *Rule Optimizer* component (H in Fig. 1); and the CritLessHunter is used variation evaluates the impact of the *Criticality Classifier* component (F in Fig. 1). When the *Criticality Classifier* was omitted (CritLessHunter variation), we observed minimal impact on entity extraction metrics. However, this component is vital for assigning appropriate threat levels and aiding in the prioritization of *Sigma* candidates. Table 12 in Appendix D presents the results for each of the variations in the previously evaluated entity and relationship identification tasks.

The results obtained with the BlindHunter variation show a 7% decrease in the F1 score for the API Call entity extraction task, with the recall dropping to 82%. Additionally, the weighted average precision and recall for *Detection Field Name ↔ Detection Entity* relationship identification were reduced by 17% and 21%, respectively. This significant reduction in accuracy, especially in extraction coverage (API Calls), highlights the importance of the *Image Classifier* and *Image Transcriptor* components in extracting information from images that may not be available elsewhere.

The NoAPIHunter variation, with the *API Call Extractor* and *TTP Extractor* components removed, resulted in significantly worse performance compared to the other variations. For the task of entity extraction, we observed a 22% drop in the average precision and a 7% drop in the average recall. Performance on the relationship extraction metrics was even more affected, with a 42% reduction in the average precision and a 14% reduction in the average recall.

These findings highlight the importance of dedicated components for entity extraction, such as the *API Call Extractor* and *TTP Classifier*, which allow the model to focus on accurate extraction before rule generation. Specifically, the *API Call Extractor* and *TTP Extractor* components proved essential to LLMCloudHunter's overall performance. In contrast, less dramatic differences in the performance were seen with the UnoptimizedHunter variation, which assesses the impact of omitting the *Rule Optimizer* component. In the relationship extraction task, there was a 17% reduction in average precision and a 9% decrease in average recall. Although these declines are not as great as those seen in the previous variation in terms of API Call extraction, the decrease in the relationship identification indicates that syntax and executability will be affected.

To summarize, the ablation study highlights the essential roles of the *Image Classifier*, *Image Transcriptor*, *API Call Extractor*, and *TTP Extractor* components in maintaining high precision and recall in both entity and relationship extraction tasks. The *Rule Optimizer* also plays a valuable role, though its impact is less pronounced compared to the other components.

## 5 DISCUSSION

Our experiments highlighted the effectiveness of various techniques applied throughout LLMCloudHunter's pipeline. These techniques, along with the purpose and specific settings for each component, are summarized in Table 13 in Appendix E and described below:
**Majority Rule in Entity Extraction Using LLMs.** We used a majority voting mechanism in the *API Call Extractor* to address LLM inconsistencies and hallucinations. While identical extraction requests generally produced similar results, occasional variations may occur due to the LLM's generative nature. To ensure accuracy, only API calls meeting a set majority threshold were retained. We experimented with the number of runs and threshold size to balance runtime, cost, and accuracy. This approach effectively reduced erroneous results in ambiguous cases.
**Structured Response Format.** For each LLM request, we use the JSON output format LLM via the request setting [24]. This structured format enables automatic validation and processing. It also allows direct access to values without additional post-processing.
**LLM Temperature Settings.** The temperature setting of an LLM influences the creativity and randomness of its outputs, and its values range between zero and two [23]. By adjusting the temperature for different tasks, we can improve the results. For example, in the *API Call Extractor* component, where extracting the information accurately is crucial, we use a low temperature of zero to ensure more accurate responses. In contrast, for the *Rule Generator* component, we set the temperature to 0.7 to allow the model to generate conditions for *Sigma* rules, which require some 'creativity.'
**Leveraging the Few-shot Learning Technique.** Providing instructions and input-output examples can significantly improve model performance [9, 26]. By dividing the OSCTI analysis into smaller tasks, we provided specific instructions for each. Using few-shot learning with a small number of examples further enhanced the model's ability to generate accurate outputs.
**Parallel LLM Requests.** We leveraged independent LLM prompts to perform parallel execution, resulting in improved speed and efficiency. We identified two key scenarios where parallel requests were particularly beneficial. First, in preprocessing, we translated all images into text simultaneously, accelerating this step. Second, in paragraph-level processing, we processed each paragraph in parallel, reducing overall processing time by threefold. This approach reduces the runtime and improves scalability for larger datasets, allowing for more efficient handling of extensive text corpora.
**Limitations.** Using a commercial LLM model (OpenAI's GPT-4o), known for its performance [5, 43], adds a cost factor that needs to be considered (approximately 25 cents per OSCTI). In addition, while we used pretrained LLMs, fine-tuning open-source models, may have an advantage in performing specific tasks correctly.

## 6 CONCLUSIONS AND FUTURE WORK

In this paper, we presented LLMCloudHunter, an end-to-end framework that analyzes textual and visual OSCTI using a pretrained LLM model when provided a URL. Our framework offers significant flexibility by allowing easy updates to newer and improved models without the need for fine-tuning, and it demonstrates scalability by running independently across multiple OSCTI images and paragraphs. By using the *Sigma* format, LLMCloudHunter's output can be seamlessly integrated into existing SIEM systems. Future work can focus on extending LLMCloudHunter to on-premise environments, increasing its applicability in diverse organizational settings and environments. Additionally, we plan to enhance our framework by equipping it with playbook automation capabilities, which will improve its ability to mitigate detected threats and provide more robust support for threat hunters.

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

# A  RELATED WORK

In this section, we provide an overview of recent studies focused on unstructured OSCTI analysis (also summarized in Table 5).

**OSCTI Analysis Techniques.** The development of efficient threat hunting mechanisms that leverage OSCTI has resulted in a wide range of research methodologies, each using different approaches to analyze and interpret OSCTI data. Within each OSCTI, key information (e.g., IoC or TTPs) is often implicit and requires the use of a different extraction approach.

NLP techniques have been utilized extensively for OSCTI analysis in methods including: Casie [37], Extractor [36], Open-CyKG [35], SecIE [28], and CyberEntRel [4]. These methods leveraged advanced NLP models (e.g., BiLSTM, BERT, RoBERTa) to extract actionable insights from OSCTI text. However, to adapt these models to the cyber threat domain, a significant amount of preprocessing and fine-tuning is required. TTPDrill [15] and THREATRAPTOR [13]

| Reference | Year | Technique | Dataset | Target Environment | Image Processing | Output | Extraction | | | | |
|---|---|---|---|---|---|---|---|---|---|---|---|
| | | | | | | | Entities | Relations | IoCs | TTPs | Detection Queries/Rules |
| TTPDrill [15] | 2017 | Unsupervised NLP | Symantec | On-premise | × | STIX | ✓ | ✓ | ✓ | ✓ | |
| Casie [37] | 2020 | BiLSTM | CyberWire | On-premise | × | Knowledge Graph | ✓ | ✓ | | ✓ | |
| Extractor [36] | 2021 | BERT-BiLSTM | APT Repository, Microsoft, Symantec, Threat Encyclopedia, Virus Radar | On-premise | × | Threat Behavior Graph | ✓ | ✓ | ✓ | | |
| Open-CyKG [35] | 2021 | BiLSTM | MalwareDB | On-premise | × | Knowledge Graph | ✓ | ✓ | | | |
| ThreatRaptor [13] | 2021 | Unsupervised NLP | DARPA TC | On-premise | × | Threat Behavior Graph, TBQL Queries | ✓ | ✓ | ✓ | ✓ | ✓ |
| SecIE [28] | 2022 | BERT | CVE | On-premise | × | Knowledge Graph | ✓ | ✓ | ✓ | | |
| CyNER [12] | 2022 | BERT | Custom | On-premise | × | STIX | ✓ | ✓ | ✓ | | |
| TriCTI [19] | 2022 | BERT | Custom | On-premise | × | Labeled IoCs | | | ✓ | ✓ | |
| LADDER [6] | 2023 | BERT | Custom | On-premise | × | Knowledge Graph | ✓ | ✓ | ✓ | ✓ | |
| Purba and Chu [29] | 2023 | GPT-3.5 | Twitter Posts | On-premise | × | Labeled IoCs | ✓ | | ✓ | | |
| aCTIon [39] | 2023 | GPT-3.5 | Custom | On-premise | × | STIX | ✓ | ✓ | ✓ | ✓ | |
| Liu and Zhan [20] | 2023 | ChatGPT | Custom | On-premise | × | Knowledge Graph | ✓ | ✓ | ✓ | | |
| LLM-TIKG [14] | 2023 | Fine-tuned LLaMA-2-7B | Custom | On-premise | × | Knowledge Graph | ✓ | ✓ | ✓ | ✓ | |
| CyberEntRel [4] | 2024 | RoBERTa-BiGRU-CRF | Custom | On-premise | × | Knowledge Graph | ✓ | ✓ | | | |
| Fengrui and Du [11] | 2024 | Fine-tuned LLaMA-2-7B | ATT&CK STIX Data | On-premise, Cloud | × | MITRE ATT&CK TTPs | | | | ✓ | |
| **Our Framework** | **2024** | **GPT-4o** | **Custom** | **Cloud** | ✓ | **Sigma Rules** | ✓ | ✓ | ✓ | ✓ | ✓ |

**Table 5: Comparison of studies utilizing OSCTI inputs.**

implement an unsupervised NLP pipeline that employs rule-based and information retrieval techniques. While this approach reduces the need for extensive model training, it is not flexible, and significant customization is needed for use in cloud environments. This is due to fundamental differences in terminology and data types between traditional on-premise environments and cloud environments, as well as the dynamic nature of cloud architectures, which continuously evolve with new services and configurations.

The introduction of LLMs has led to a paradigm shift in OSCTI processing, with research demonstrating their ability to extract meaningful and structured data from OSCTI text. Utilizing GPT-3.5, Purba and Chu [29] and Siracusano et al. [39] addressed tasks ranging from the extraction of IoCs to the generation of structured CTI format (e.g., STIX), respectively, while Liu and Zhan [20] applied ChatGPT to construct graphical representations of OSCTI data. Hu et al. [14] and Fengrui and Du [11] expanded upon these capabilities by utilizing both pretrained and fine-tuned LLM models. They employed GPT-3.5 and ChatGPT for data annotation and augmentation, respectively, to prepare datasets for fine-tuning the LLaMA2-7B model. Hu et al. [14] applied the fine-tuned LLaMA2-7B to construct knowledge graphs, while Fengrui and Du [11] focused on TTP classification. In this research, we are the first to develop an end-to-end framework based on a pretrained LLM, demonstrating the potential of LLMs in processing OSCTI and generating actionable *Sigma* rules. Moreover, our framework integrates visual analysis capabilities, expanding the scope of OSCTI analysis beyond previous text-centric methodologies. By leveraging pretrained LLMs, we avoid the need for rule-based methods or training customized models with dedicated datasets. Our framework also focuses on generating rules for cloud environments, which has not been addressed before.

**Datasets.** In terms of OSCTI datasets, the study introducing TTPDrill [15] used a dataset of semi-structured Symantec threat reports, from which threat actions were manually extracted. Similarly, Satyapanich et al. [37] employed cybersecurity news articles published on CyberWire,[2] which were annotated before evaluation. The study presenting Extractor [36] used multiple structured OSCTI sources, including Microsoft, Symantec, Threat Encyclopedia, and Virus Radar. Open-CyKG [35] used a structured OSCTI database focusing on malware. ThreatRaptor [13] utilized the DARPA TC dataset, incorporating semi-structured OSCTIs, along with IoCs and relevant event log entries for each attack incident. SecIE [28] used 133 unstructured labeled threat reports from various threat intelligence vendors. CyNer [12] and TriCTI [19] developed a custom web crawler to retrieve unstructured OSCTIs across selected high-quality websites (e.g., Kaspersky, Symantec, and Fireye) and manually annotated a subset for evaluation purposes. LLM-TIKG [14] also developed a custom web crawler to collect OSCTIs from selected platforms, but this study differs from the study presenting CyNer in that it utilizes an LLM (GPT) for annotation. Liu and Zhan [20] manually collected OSCTIs from public sites, and for each OSCTI, they selected the paragraphs that refer to the target technique to increase information density. LADDER [6] used OSCTI reports related to a specific set of malware, employing the BRAT [40] NLP method to annotate the concepts and their relationships in the text. Purba and Chu [29] analyzed a dataset comprising 150 cyber threat related tweets. aCTIon [39] manually collected OSCTI posts and their respective STIX bundles and used expert-based annotation to create the ground truth. CyberEntRel [4] collected OSCTI reports from high-quality vendors (e.g., Microsoft, Cisco, McAfee, and Kaspersky) and performed keyword-based data extraction. Fengrui and Du [11] used the MITRE ATT&CK dataset, structured in STIX 2.1 JSON format, which is a tagged and organized collection of adversary tactics and techniques.

In contrast to prior studies that primarily used semi-structured and on-premise-related datasets, we use 20 unstructured, publicly available *cloud-based* posts and reports sourced from various publishers. These OSCTI reports, which describe AWS cloud incidents,

---

[2]https://thecyberwire.com/

were systematically annotated by our research team to develop a robust ground truth for development and evaluation.

**Extractions and Outputs.** Previous studies produced a variety of outputs with different levels of utility and applicability. Liu et al. [19] and Purba and Chu [29] focused on extracting IoCs, while Fengrui and Du [11] extracted TTPs. The studies presenting TTP-Drill [15], Cyner [12], and aCTIon [39] converted unstructured OSCTIs into structured representations using the STIX format, which facilitates the systematic sharing and analysis of threat information. A more advanced approach was used in Extractor [37] and ThreatRaptor [13], in which threat behavior graphs are created; and in Casie [37], Open-CyKG [35], SecIE [28], LADDER [6], aCTIon [20], LLM-TIKG [14], CyberEntRel [4], in which knowledge graphs are generated. Both approaches interrelate entities with associated actions and artifacts (i.e., IoCs and TTPs), providing structured insights into attack strategies through graph-based representations. While the approaches highlighted above provide valuable contextual information, further processing is required to transform the representations into actionable defense mechanisms.

To address this, in their study, Gao et al. presented a framework for converting OSCTI data into a threat behavior graph and associated domain-specific queries. Both frameworks go beyond simply identifying and contextualizing threat data, by developing operational detection rules and queries. The detection rule candidates generated by LLMCloudHunter, however, are in the known open-source *Sigma* structure. This widely used generic signature format is inherently suitable for integration in various application environments and SIEMs. By capturing the entities, relations, IoCs, and TTPs identified in OSCTI, LLMCloudHunter translates threat intelligence into applicative *Sigma* candidates.

## B  OSCTI SOURCES USED IN OUR RESEARCH

Table 6 presents the list of OSCTI sources used in the development and evaluation of LLMCloudHunter. For each source, we provide the number of images included, the number of tokens (which serve as input to the LLM), the number of API calls, and our rating of the OSCTI's technical complexity. The complete results of the entity and relationship extraction are presented in the following tables: detection entities and relationships in Tables 7 and 8, and MITRE ATT&CK entities and relationships in Tables 9 and 10.

## C  RUNNING EXAMPLE

This section provides a step-by-step demonstration of the LLMCloudHunter framework in action. The example uses an actual OSCTI source, specifically a Sysdig blog post titled *"SCARLETEEL: Operation leveraging Terraform, Kubernetes, and AWS for data theft"*[3], which describes a cloud infrastructure exploit. This demonstration focuses on specific paragraphs relevant to the *Sigma* rule being generated, leading to the rule presented in Listing 1.

**Step 1 (*Preprocessing* Phase).** The initial phase involves pre-processing unstructured OSCTI data, which can be seen in Fig. 2a. In this phase, the website content is downloaded and parsed by the *Downloader and Parser* component, which converts HTML code into a markdown format. The *Image Analyzer* component processes

---

the embedded images to extract relevant text. This phase results in the formatted textual output shown in Fig. 2b.

**Step 2 (*Paragraph-Level Processing* Phase).** In this phase, we first extract API calls and then classify each one according to its corresponding MITRE ATT&CK TTPs and criticality level. These extractions are then attached to the formatted paragraph content to enrich it with additional context. Fig. 3 demonstrates how the API calls *'ListBuckets'* and *'GetObject'*, along with their event sources and TTPs, are added to the output in our example. This enriched paragraph is then fed to the *Rule Generator* component to generate an initial *Sigma* rule, as shown in Listing 2.

```
title: Access to Terraform File
description: Detects requests for terraform.tfstate file.
  This file contains sensitive infrastructure information
  and secrets, indicating potential compromise or
  unauthorized access.
references:
    - https://sysdig.com/blog/cloud-breach-terraform-data-
  theft/
    - https://docs.aws.amazon.com/AmazonS3/latest/API/
  API_GetObject.html
author: LLMCloudHunter
tags:
    - attack.collection
    - attack.t1530
logsource:
    product: aws
    service: cloudtrail
detection:
    selection_event:
        eventSource: s3.amazonaws.com
        eventName: GetObject
        requestParameters.key: terraform.tfstate
    condition: selection_event
falsepositives:
  - Automated CI/CD pipeline operations
  - DevOps engineers manually running Terraform commands
level: high
```

**Listing 2: The initial generated Sigma rule.**

**Step 3 (*OSCTI-Level Processing* Phase).** The *Rule Optimizer* component refines the detection logic of each rule. In our case, it finds no faults to fix and leaves the initial rule as it is. The *Set Refiner* removes duplicates in the overall set; here, there is no duplication of the *'GetObject'* API call. The *IoC Enhancer* then uses the extracted IoCs in the IoC paragraph (Fig. 2) to enhance the rule with suspicious IP addresses. This results in the final *Sigma* rule presented in Listing 1.

## D  ABLATION STUDY

In this section, we present the configurations used in the ablation study and our results. Table 11 lists the different configurations, indicating which components were used in each variation.

Table 12 contains the results of the experiments performed in the ablation study, presenting the weighted average precision, recall, and F1 score for each variation and extraction type.

## E  COMPONENT CONFIGURATIONS

In this section, we provide details on the configuration of each component in the LLMCloudHunter framework. For each component, Table 13 lists the purpose, techniques, and parameters.

**Credential access – Terraform state files**

Terraform is an open source infrastructure as code (IaC) tool used to deploy, change, or create infrastructures in cloud environments.

In order for Terraform to know which resources are under its control and when to update and destroy them, it uses a state file named *terraform.tfstate* by default. When Terraform is integrated and automated in continuous integration/continuous delivery (CI/CD) pipelines, the state file needs to be accessible with proper permissions. In particular, the service principal running the pipeline needs to be able to access the storage account container that holds the state file. This makes shared storage like Amazon S3 buckets a perfect candidate to hold the state file.

However, Terraform state files contain all data in plain text, which may contain secrets. Storing secrets anywhere other than a secure location is never a good idea, and definitely should not be put into source control!

The attacker was able to list the bucket available and retrieve all of the data. Examining the data with different tools such as Pacu and TruffleHog during the incident investigation, it was possible to find both a clear-text IAM user access key and secret key in the *terraform.tfstate* file inside of an S3 bucket. Here is a screenshot from TruffleHog.

```
Found verified result 🐷🔑
Detector Type: AWS
Decoder Type: PLAIN
Raw result: AKIA2
Bucket:
Email:
File:                    terraform/terraform.tfstate
Link: https://           terraform/terraform.tfstate
```

These IAM credentials are for a second connected AWS account, giving the attacker the opportunity to move laterally to spread their attack throughout the organization.

**IoCs**

**IP Addresses:**

- 80[.]239[.]140[.]66
- 45[.]9[.]148[.]221
- 45[.]9[.]148[.]121
- 45[.]9[.]249[.]58

---

```
## Credential access – Terraform state files

Terraform is an open source infrastructure as code (IaC) tool used to deploy, change,
or create infrastructures in cloud environments.
In order for Terraform to know which resources are under its control and when
to update and destroy them, it uses a state file named terraform.tfstate by default. When
Terraform is integrated and automated in continuous integration/continuous delivery (CI/CD)
pipelines, the state file needs to be accessible with proper permissions. In particular, the
service principal running the pipeline needs to be able to access the storage account
container that holds the state file. This makes shared storage like Amazon S3 buckets a
perfect candidate to hold the state file.
However, Terraform state files contain all data in plain text, which may contain secrets.
Storing secrets anywhere other than a secure location is never a good idea, and definitely
should not be put into source control!
The attacker was able to list the bucket available and retrieve all of the data. Examining
the data with different tools such as Pacu and TruffleHog during the incident investigation,
it was possible to find both a clear-text IAM user access key and secret key in the
terraform.tfstate file inside of an S3 bucket. Here is a screenshot from TruffleHog.

[Image Info:
- Alt Text: Terraform s3 bucket leak credentials
- Description: The image shows a screenshot of a command line interface output related to a
cybersecurity investigation or monitoring tool.
- Trancription:
*Found verified result 🐷🔑*
*Detector Type:* AWS
*Decoder Type:* PLAIN
*Raw result:* AKIA2
*Bucket:* [Obscured]
*Email:* [Obscured]
*File:* terraform/terraform.tfstate
*Link:* https://[Obscured]/terraform/terraform.tfstate

From the details above:
- The "Bucket" might be an Amazon S3 bucket which is a part of AWS (Amazon Web Services).
- The "Raw result" starting with "AKIA2" suggests the presence of an AWS Access Key ID.
- The link indicated ("https:/ /terraform/terraform.tfstate") suggests that Terraform, an
infrastructure as code software tool, is being used, and the specific file mentioned
is "terraform.tfstate," a file used by Terraform to store state data which can
include sensitive information.]

These IAM credentials are for a second connected AWS account, giving the attacker the
opportunity to move laterally to spread their attack throughout the organization.

## IoCs

### IP Addresses:

- 80[.]239[.]140[.]66
- 45[.]9[.]148[.]221
- 45[.]9[.]148[.]221
- 45[.]9[.]249[.]58
```

(a) Screenshots of two paragraphs from the OSCTI    (b) Corresponding preprocessed output

**Figure 2: OSCTI Preprocessing phase.**

```
CTI Paragraph: """
preprocessed_paragraph_here
"""

Identified API Calls: """
[
    {
        "eventName": "ListBuckets",
        "eventSource": "s3.amazonaws.com",
        "tags": {
            "tactic_name": "attack.discovery",
            "technique_id": "attack.t1580"
        },
        "level": "low"
    },
    {
        "eventName": "GetObject",
        "eventSource": "s3.amazonaws.com",
        "tags": {
            "tactic_name": "attack.collection",
            "technique_id": "attack.t1530"
        },
        "level": "medium"
    }
]
"""
```

**Figure 3: Formatted and enriched paragraph (input for the Rule Generator component).**

| OSCTI ID | OSCTI Name | #Images | #Tokens | | #API Calls | Technical Complexity |
|---|---|---|---|---|---|---|
| | | | No Images | Images | | |
| 1 | Anatomy of an Attack: Exposed keys to Crypto Mining | 1 | 1254 | 1511 | 11 | High |
| 2 | Behind the scenes in the Expel SOC: Alert-to-fix in AWS | 7 | 3136 | 4892 | 11 | Medium |
| 3 | Bling Libra's Tactical Evolution: The Threat Actor Group Behind ShinyHunters Ransomware | 20 | 6414 | 11391 | 20 | High |
| 4 | CloudKeys in the Air: Tracking Malicious Operations of Exposed IAM Keys | 10 | 5792 | 10884 | 21 | Low |
| 5 | Compromised Cloud Compute Credentials: Case Studies From the Wild (Case 1) | 1 | 2448 | 2718 | 51 | Low |
| 6 | Detecting AI resource-hijacking with Composite Alerts | 4 | 2952 | 4078 | 22 | Medium |
| 7 | Finding evil in AWS: A key pair to remember | 7 | 2852 | 3814 | 11 | Medium |
| 8 | Incident report: From CLI to console, chasing an attacker in AWS | 1 | 2326 | 3504 | 11 | Medium |
| 9 | Incident report: stolen AWS access keys | 4 | 1984 | 3998 | 7 | Medium |
| 10 | LUCR-3: Scattered Spider Getting SaaS-y in the Cloud | 2 | 3666 | 4143 | 20 | Low |
| 11 | Ransomware in the cloud | 7 | 4743 | 5931 | 17 | High |
| 12 | SCARLETEEEL: Operation leveraging Terraform, Kubernetes, and AWS for data theft | 12 | 3671 | 9764 | 26 | Medium |
| 13 | Tales from the cloud trenches: Amazon ECS is the new EC2 for crypto mining | 2 | 4784 | 5209 | 23 | Medium |
| 14 | Tales from the cloud trenches: Raiding for AWS vaults, buckets and secrets | 2 | 2027 | 2310 | 9 | Medium |
| 15 | Tales from the cloud trenches: Using AWS CloudTrail to identify malicious activity and spot phishing campaign | 8 | 3602 | 5187 | 6 | Medium |
| 16 | The curious case of DangerDev@protonmail.me | 31 | 7541 | 14465 | 60 | Medium |
| 17 | Two real-life examples of why limiting permissions works: Lessons from AWS CIRT (Case 1) | 0 | 2160 | 2160 | 9 | Low |
| 18 | Two real-life examples of why limiting permissions works: Lessons from AWS CIRT (Case 2) | 0 | 2059 | 2059 | 7 | Low |
| 19 | Unmasking GUI-Vil: Financially Motivated Cloud Threat Actor | 7 | 7604 | 9018 | 13 | High |
| 20 | When a Zero Day and Access Keys Collide in the Cloud: Responding to the SugarCRM Zero-Day Vulnerability | 6 | 4922 | 5743 | 20 | High |

**Table 6: OSCTI sources used in our research.**

# F  PROMPTS

In this section, we provide the various prompts utilized throughout our proposed method in the LLMCloudHunter framework. The prompts associated with each component are described below:

- **API Call Extractor** (D in Fig. 1): This component extracts explicit and implicit API calls from the dataset. The methodology employed in this component is illustrated in Fig. 4. The explicit and implicit API call extraction prompts are shown in Figures 5 and 6, respectively. Additionally, the prompts used for the image classification and transcription sub-components are provided in Figures 7 and 8.

- **TTP Classifier** (E in Fig. 1): The prompt used for classifying threat tactics, techniques, and procedures (TTPs) is detailed in Figure 9.
- **Criticality Classifier** (F in Fig. 1): The prompt used to evaluate the criticality of specific elements is shown in Figure 10.
- **Rule Generator** (G in Fig. 1): The prompt used for generating *Sigma* rules is provided in Figure 11.
- **Rule Optimizer** (H in Fig. 1): The prompt used for the rule optimization process is outlined in Figure 12.
- **Rule Selector** (J in Fig. 1): The prompt for selecting the most suitable rules is shown in Figure 13.

| OSCTI ID | Detection Field Name | | | | Log Source | | | | API Call | | | | IoC | | | | Other | | | |
|---|---|---|---|---|---|---|---|---|---|---|---|---|---|---|---|---|---|---|---|---|
| | Support | Precision | Recall | F1-score | Support | Precision | Recall | F1-score | Support | Precision | Recall | F1-score | Support | Precision | Recall | F1-score | Support | Precision | Recall | F1-score |
| 1 | 16 | 0.6 | 0.38 | 0.46 | 2 | 1 | 1 | 1 | 11 | 0.85 | 1 | 0.92 | 2 | 0.67 | 1 | 0.8 | 9 | 1 | 0.22 | 0.36 |
| 2 | 8 | 1 | 0.88 | 0.93 | 2 | 1 | 1 | 1 | 11 | 0.92 | 1 | 0.96 | 3 | 1 | 1 | 1 | 2 | 1 | 0.5 | 0.67 |
| 3 | 8 | 0.86 | 0.75 | 0.8 | 2 | 1 | 1 | 1 | 20 | 0.74 | 1 | 0.85 | 5 | 1 | 0.6 | 0.75 | 2 | 0 | 0 | 0 |
| 4 | 7 | 1 | 0.86 | 0.92 | 2 | 1 | 1 | 1 | 21 | 1 | 0.95 | 0.98 | 2 | 1 | 1 | 1 | 9 | 1 | 0.56 | 0.71 |
| 5 | 6 | 1 | 0.83 | 0.91 | 2 | 1 | 1 | 1 | 51 | 0.82 | 1 | 0.9 | 1 | 1 | 1 | 1 | 2 | 0 | 0 | 0 |
| 6 | 6 | 1 | 1 | 1 | 2 | 1 | 1 | 1 | 22 | 0.85 | 1 | 0.92 | 50 | 1 | 1 | 1 | 0 | 1 | 1 | 1 |
| 7 | 6 | 0.75 | 1 | 0.86 | 2 | 1 | 1 | 1 | 11 | 0.85 | 1 | 0.92 | 2 | 1 | 1 | 1 | 0 | 1 | 1 | 1 |
| 8 | 11 | 1 | 0.73 | 0.84 | 2 | 1 | 1 | 1 | 11 | 0.92 | 1 | 0.96 | 5 | 1 | 1 | 1 | 4 | 1 | 0.25 | 0.4 |
| 9 | 5 | 1 | 1 | 1 | 2 | 1 | 1 | 1 | 7 | 1 | 1 | 1 | 3 | 0.75 | 1 | 0.86 | 0 | 1 | 1 | 1 |
| 10 | 6 | 1 | 0.83 | 0.91 | 2 | 1 | 1 | 1 | 20 | 0.74 | 1 | 0.85 | 3 | 1 | 0.67 | 0.8 | 1 | 0 | 0 | 0 |
| 11 | 7 | 0.88 | 1 | 0.93 | 2 | 1 | 1 | 1 | 17 | 0.89 | 1 | 0.94 | 67 | 1 | 1 | 1 | 5 | 0.83 | 1 | 0.91 |
| 12 | 6 | 1 | 0.83 | 0.91 | 2 | 1 | 1 | 1 | 26 | 1 | 0.96 | 0.98 | 4 | 1 | 1 | 1 | 1 | 0 | 0 | 0 |
| 13 | 10 | 0.5 | 1 | 0.67 | 2 | 1 | 1 | 1 | 23 | 0.66 | 1 | 0.79 | 0 | 1 | 1 | 1 | 4 | 0.31 | 1 | 0.47 |
| 14 | 6 | 0.86 | 1 | 0.92 | 2 | 1 | 1 | 1 | 9 | 0.64 | 1 | 0.78 | 10 | 1 | 1 | 1 | 0 | 1 | 1 | 1 |
| 15 | 6 | 1 | 0.83 | 0.91 | 2 | 1 | 1 | 1 | 6 | 1 | 1 | 1 | 2 | 1 | 1 | 1 | 1 | 0 | 0 | 0 |
| 16 | 15 | 0.72 | 0.87 | 0.79 | 2 | 1 | 1 | 1 | 60 | 0.77 | 1 | 0.87 | 8 | 1 | 1 | 1 | 17 | 0.7 | 0.82 | 0.76 |
| 17 | 6 | 0.83 | 0.83 | 0.83 | 2 | 1 | 1 | 1 | 9 | 0.64 | 1 | 0.78 | 0 | 1 | 1 | 1 | 2 | 1 | 0.5 | 0.67 |
| 18 | 6 | 0.83 | 0.83 | 0.83 | 2 | 1 | 1 | 1 | 7 | 0.78 | 1 | 0.88 | 0 | 1 | 1 | 1 | 3 | 1 | 0.67 | 0.8 |
| 19 | 11 | 0.92 | 1 | 0.96 | 2 | 1 | 1 | 1 | 13 | 0.81 | 1 | 0.9 | 10 | 1 | 1 | 1 | 3 | 0.75 | 1 | 0.86 |
| 20 | 12 | 0.86 | 1 | 0.92 | 2 | 1 | 1 | 1 | 20 | 0.86 | 0.95 | 0.9 | 13 | 1 | 0.77 | 0.87 | 4 | 0.8 | 1 | 0.89 |
| Weighted Average | 8.20 | 0.85 | 0.85 | 0.84 | 2.00 | 1.00 | 1.00 | 1.00 | 18.75 | 0.83 | 0.99 | 0.90 | 9.50 | 0.99 | 0.97 | 0.98 | 3.45 | 0.75 | 0.61 | 0.61 |

Table 7: Detection entity results.

| OSCTI ID | Detection Field Name ↔ Detection Entity | | | | API Call ↔ API Source | | | | API Call ↔ Log Source | | | | API Call ↔ IoC | | | | API Call ↔ Other | | | |
|---|---|---|---|---|---|---|---|---|---|---|---|---|---|---|---|---|---|---|---|---|
| | Support | Precision | Recall | F1-score | Support | Precision | Recall | F1-score | Support | Precision | Recall | F1-score | Support | Precision | Recall | F1-score | Support | Precision | Recall | F1-score |
| 1 | 24 | 1 | 0.62 | 0.77 | 8 | 1 | 1 | 1 | 16 | 1 | 1 | 1 | 15 | 0.94 | 1 | 0.97 | 9 | 1 | 0.22 | 0.36 |
| 2 | 18 | 1 | 0.94 | 0.97 | 9 | 1 | 1 | 1 | 18 | 1 | 1 | 1 | 27 | 1 | 1 | 1 | 7 | 1 | 0.14 | 0.25 |
| 3 | 29 | 1 | 0.86 | 0.93 | 12 | 1 | 1 | 1 | 36 | 1 | 1 | 1 | 54 | 1 | 1 | 1 | 2 | 0 | 0 | 0 |
| 4 | 34 | 1 | 0.85 | 0.92 | 8 | 1 | 1 | 1 | 32 | 1 | 0.94 | 0.97 | 18 | 1 | 1 | 1 | 9 | 1 | 0.56 | 0.71 |
| 5 | 56 | 1 | 0.96 | 0.98 | 72 | 1 | 0.6 | 0.75 | 90 | 1 | 1 | 1 | 45 | 1 | 1 | 1 | 4 | 0 | 0 | 0 |
| 6 | 87 | 1 | 0.95 | 0.98 | 60 | 1 | 0.83 | 0.91 | 96 | 1 | 1 | 1 | 384 | 1 | 1 | 1 | 20 | 1 | 0.8 | 0.89 |
| 7 | 15 | 1 | 1 | 1 | 14 | 1 | 0.71 | 0.83 | 16 | 1 | 1 | 1 | 14 | 1 | 1 | 1 | 0 | 1 | 1 | 1 |
| 8 | 22 | 1 | 0.82 | 0.9 | 8 | 1 | 1 | 1 | 16 | 1 | 1 | 1 | 13 | 0.52 | 1 | 0.68 | 4 | 1 | 0.25 | 0.4 |
| 9 | 12 | 0.92 | 1 | 0.96 | 4 | 1 | 1 | 1 | 8 | 1 | 1 | 1 | 12 | 0.8 | 1 | 0.89 | 0 | 1 | 1 | 1 |
| 10 | 26 | 1 | 0.92 | 0.96 | 14 | 1 | 1 | 1 | 28 | 1 | 1 | 1 | 23 | 1 | 0.96 | 0.98 | 1 | 0 | 0 | 0 |
| 11 | 91 | 1 | 1 | 1 | 13 | 0.92 | 0.92 | 0.92 | 26 | 1 | 1 | 1 | 804 | 0.99 | 1 | 0.99 | 5 | 1 | 1 | 1 |
| 12 | 6 | 1 | 0.83 | 0.91 | 2 | 1 | 1 | 1 | 26 | 1 | 0.96 | 0.98 | 4 | 1 | 1 | 1 | 1 | 0 | 0 | 0 |
| 13 | 29 | 1 | 1 | 1 | 16 | 1 | 1 | 1 | 32 | 1 | 1 | 1 | 0 | 1 | 1 | 1 | 4 | 1 | 1 | 1 |
| 14 | 21 | 1 | 1 | 1 | 13 | 1 | 0.62 | 0.76 | 10 | 1 | 1 | 1 | 37 | 0.82 | 1 | 0.9 | 0 | 1 | 1 | 1 |
| 15 | 11 | 1 | 0.91 | 0.95 | 4 | 1 | 1 | 1 | 8 | 1 | 1 | 1 | 6 | 1 | 1 | 1 | 2 | 0 | 0 | 0 |
| 16 | 87 | 1 | 0.95 | 0.98 | 60 | 1 | 0.83 | 0.91 | 96 | 1 | 1 | 1 | 384 | 1 | 1 | 1 | 20 | 1 | 0.8 | 0.89 |
| 17 | 13 | 1 | 0.92 | 0.96 | 8 | 1 | 1 | 1 | 16 | 1 | 1 | 1 | 0 | 1 | 1 | 1 | 16 | 1 | 0.38 | 0.55 |
| 18 | 12 | 1 | 0.75 | 0.86 | 4 | 1 | 0.5 | 0.67 | 8 | 1 | 0.5 | 0.67 | 0 | 1 | 1 | 1 | 7 | 1 | 0.29 | 0.44 |
| 19 | 28 | 1 | 1 | 1 | 9 | 1 | 1 | 1 | 18 | 1 | 1 | 1 | 90 | 1 | 1 | 1 | 3 | 1 | 1 | 1 |
| 20 | 39 | 1 | 0.97 | 0.99 | 14 | 1 | 0.93 | 0.96 | 28 | 1 | 0.93 | 0.96 | 104 | 0.77 | 0.92 | 0.84 | 4 | 1 | 1 | 1 |
| Weighted Average | 33.00 | 1.00 | 0.93 | 0.97 | 17.60 | 1.00 | 0.82 | 0.89 | 31.20 | 1.00 | 0.99 | 0.99 | 101.70 | 0.98 | 1.00 | 0.98 | 5.90 | 0.92 | 0.55 | 0.65 |

Table 8: Detection relationships results.

| OSCTI ID | Technique | | | | Sub-Technique | | | |
|---|---|---|---|---|---|---|---|---|
| | Support | Precision | Recall | F1-score | Support | Precision | Recall | F1-score |
| 1 | 5 | 0.83 | 1 | 0.91 | 4 | 0.8 | 1 | 0.89 |
| 2 | 6 | 0.83 | 0.83 | 0.83 | 3 | 0.67 | 0.67 | 0.67 |
| 3 | 8 | 0.64 | 0.88 | 0.74 | 3 | 0.6 | 1 | 0.75 |
| 4 | 8 | 0.71 | 0.62 | 0.67 | 2 | 0.67 | 1 | 0.8 |
| 5 | 5 | 0.27 | 0.6 | 0.37 | 1 | 0.2 | 1 | 0.33 |
| 6 | 7 | 0.67 | 0.86 | 0.75 | 3 | 0.75 | 1 | 0.86 |
| 7 | 4 | 0.2 | 0.25 | 0.22 | 0 | 0 | 0 | 0 |
| 8 | 5 | 0.83 | 1 | 0.91 | 4 | 0.8 | 1 | 0.89 |
| 9 | 3 | 1 | 1 | 1 | 1 | 1 | 1 | 1 |
| 10 | 6 | 0.67 | 1 | 0.8 | 8 | 1 | 1 | 1 |
| 11 | 6 | 0.67 | 1 | 0.8 | 2 | 0.5 | 1 | 0.67 |
| 12 | 9 | 0.78 | 0.78 | 0.78 | 6 | 1 | 0.83 | 0.91 |
| 13 | 11 | 0.85 | 1 | 0.92 | 6 | 1 | 1 | 1 |
| 14 | 3 | 0.75 | 1 | 0.86 | 1 | 1 | 1 | 1 |
| 15 | 4 | 0.67 | 0.5 | 0.57 | 2 | 0 | 0 | 0 |
| 16 | 14 | 0.76 | 0.93 | 0.84 | 7 | 0.58 | 1 | 0.74 |
| 17 | 4 | 0.17 | 0.25 | 0.2 | 0 | 0 | 0 | 0 |
| 18 | 5 | 0.67 | 0.8 | 0.73 | 0 | 0 | 0 | 0 |
| 19 | 6 | 0.5 | 0.67 | 0.57 | 5 | 0.43 | 0.6 | 0.5 |
| 20 | 6 | 0.5 | 0.83 | 0.62 | 2 | 0.25 | 1 | 0.4 |
| Weighted Average | 6.25 | 0.67 | 0.82 | 0.73 | 3.00 | 0.73 | 0.90 | 0.79 |

Table 9: MITRE entity results.

- **API Call Remover** (K in Fig. 1): The prompt used to refine detection accuracy by removing redundant API calls is illustrated in Figure 14.

- **IoC Extractor** (L in Fig. 1): The prompt for extracting indicators of compromise (IoCs) is displayed in Figure 15.

| OSCTI ID | API Call ↔ Technique | | | | API Call ↔ Sub-technique | | | |
|---|---|---|---|---|---|---|---|---|
| | Support | Precision | Recall | F1-score | Support | Precision | Recall | F1-score |
| 1 | 8 | 0.88 | 0.88 | 0.88 | 6 | 1 | 0.83 | 0.91 |
| 2 | 9 | 1 | 0.89 | 0.94 | 4 | 1 | 0.75 | 0.86 |
| 3 | 25 | 0.91 | 0.84 | 0.87 | 6 | 1 | 0.83 | 0.91 |
| 4 | 17 | 0.54 | 0.41 | 0.47 | 4 | 1 | 0.75 | 0.86 |
| 5 | 45 | 0.5 | 0.07 | 0.12 | 2 | 1 | 1 | 1 |
| 6 | 54 | 0.4 | 0.35 | 0.38 | 13 | 0.89 | 0.62 | 0.73 |
| 7 | 10 | 0 | 0 | 0 | 0 | 1 | 1 | 1 |
| 8 | 8 | 0.88 | 0.88 | 0.88 | 6 | 1 | 0.67 | 0.8 |
| 9 | 4 | 1 | 1 | 1 | 2 | 1 | 1 | 1 |
| 10 | 14 | 1 | 1 | 1 | 12 | 0.75 | 0.75 | 0.75 |
| 11 | 12 | 0.9 | 0.75 | 0.82 | 6 | 1 | 0.83 | 0.91 |
| 12 | 9 | 0.78 | 0.78 | 0.78 | 6 | 1 | 0.83 | 0.91 |
| 13 | 16 | 0.87 | 0.81 | 0.84 | 7 | 1 | 0.86 | 0.92 |
| 14 | 9 | 1 | 0.89 | 0.94 | 1 | 1 | 1 | 1 |
| 15 | 5 | 0.67 | 0.4 | 0.5 | 2 | 0 | 0 | 0 |
| 16 | 54 | 0.4 | 0.35 | 0.38 | 13 | 0.89 | 0.62 | 0.73 |
| 17 | 10 | 0.25 | 0.1 | 0.14 | 0 | 1 | 1 | 1 |
| 18 | 7 | 1 | 0.29 | 0.44 | 0 | 1 | 1 | 1 |
| 19 | 13 | 0.71 | 0.38 | 0.5 | 11 | 1 | 0.27 | 0.43 |
| 20 | 8 | 0.5 | 0.5 | 0.5 | 2 | 1 | 1 | 1 |
| Weighted Average | 16.85 | 0.61 | 0.47 | 0.51 | 5.15 | 0.92 | 0.69 | 0.77 |

Table 10: MITRE relationships results.

| | BlindHunter | NoAPIHunter | UnoptimizedHunter | CritLessHunter | LLMCloudHunter |
|---|---|---|---|---|---|
| Downloader and Parser (A) | ✓ | ✓ | ✓ | ✓ | ✓ |
| Image Classifier (B) | | ✓ | ✓ | ✓ | ✓ |
| Image Transcriptor (C) | | ✓ | ✓ | ✓ | ✓ |
| API Call Extractor (D) | ✓ | | ✓ | ✓ | ✓ |
| TTP Classifier (E) | | ✓ | ✓ | ✓ | ✓ |
| Criticality Classifier (F) | | | | | ✓ |
| Rule Generator (G) | ✓ | ✓ | ✓ | ✓ | ✓ |
| Rule Optimizer (H) | ✓ | ✓ | | ✓ | ✓ |
| Rule Deduplicator (I) | ✓ | ✓ | ✓ | ✓ | ✓ |
| Rule Selector (J) | ✓ | ✓ | ✓ | ✓ | ✓ |
| API Call Remover (K) | ✓ | ✓ | ✓ | ✓ | ✓ |
| IoC Extractor (L) | ✓ | ✓ | ✓ | ✓ | ✓ |
| IoC Enhancer (M) | ✓ | ✓ | ✓ | ✓ | ✓ |

Table 11: Ablation study configurations.

| Extraction | Weighted Average Measure | LLMCloudHunter | CritLessHunter | BlindHunter | NoAPIHunter | UnoptimizedHunter |
|---|---|---|---|---|---|---|
| API Call | Precision | 0.83 | 0.88 | 0.85 | 0.61 | 0.88 |
| | Recall | 0.99 | 0.95 | 0.82 | 0.92 | 0.92 |
| | F1 Score | 0.90 | 0.91 | 0.83 | 0.73 | 0.90 |
| Technique | Precision | 0.67 | 0.62 | 0.58 | 0.24 | 0.57 |
| | Recall | 0.82 | 0.75 | 0.59 | 0.27 | 0.62 |
| | F1 Score | 0.73 | 0.68 | 0.57 | 0.24 | 0.58 |
| Sub-technique | Precision | 0.73 | 0.65 | 0.50 | 0.29 | 0.63 |
| | Recall | 0.90 | 0.71 | 0.53 | 0.24 | 0.64 |
| | F1 Score | 0.79 | 0.67 | 0.50 | 0.25 | 0.62 |
| IoC | Precision | 0.99 | 0.99 | 0.93 | 0.96 | 0.96 |
| | Recall | 0.97 | 0.98 | 0.90 | 0.98 | 0.97 |
| | F1 Score | 0.98 | 0.98 | 0.90 | 0.97 | 0.96 |
| Other | Precision | 0.75 | 0.77 | 0.74 | 0.37 | 0.75 |
| | Recall | 0.61 | 0.70 | 0.51 | 0.47 | 0.67 |
| | F1 Score | 0.67 | 0.73 | 0.56 | 0.39 | 0.69 |
| Detection Field Name ↔ Detection Entity | Precision | 1.00 | 0.87 | 0.83 | 0.58 | 0.83 |
| | Recall | 0.93 | 0.90 | 0.72 | 0.79 | 0.84 |
| | F1 Score | 0.97 | 0.88 | 0.75 | 0.65 | 0.83 |
| API Call ↔ Technique | Precision | 0.61 | 0.56 | 0.44 | 0.10 | 0.46 |
| | Recall | 0.47 | 0.64 | 0.42 | 0.13 | 0.49 |
| | F1 Score | 0.51 | 0.59 | 0.42 | 0.11 | 0.47 |
| API Call ↔ Sub-technique | Precision | 0.92 | 0.56 | 0.36 | 0.09 | 0.53 |
| | Recall | 0.69 | 0.58 | 0.35 | 0.11 | 0.54 |
| | F1 Score | 0.77 | 0.56 | 0.35 | 0.09 | 0.51 |
| API Call ↔ IoC | Precision | 0.98 | 0.92 | 0.94 | 0.70 | 0.88 |
| | Recall | 1.00 | 0.91 | 0.75 | 0.93 | 0.87 |
| | F1 Score | 0.98 | 0.92 | 0.83 | 0.78 | 0.87 |
| API Call ↔ Other | Precision | 0.92 | 0.84 | 0.74 | 0.41 | 0.72 |
| | Recall | 0.55 | 0.85 | 0.61 | 0.41 | 0.76 |
| | F1 Score | 0.65 | 0.84 | 0.65 | 0.39 | 0.72 |

Table 12: Ablation study results.

| Component | Purpose | LLM Utilization | Structured Response | Leverage Few-Shot | Temperature | Parallel Requests |
|---|---|---|---|---|---|---|
| A | HTML downloading and parsing | | | | | |
| B | Image Classification | ✓ | ✓ | | 1 | ✓ |
| C | Image Transcription | ✓ | ✓ | | 1 | ✓ |
| D | Explicit API call extracting | ✓ | ✓ | | 0 | ✓ |
| | Implicit API call extracting | ✓ | ✓ | ✓ | 0.9 | ✓ |
| E | TTPs extracting | ✓ | ✓ | ✓ | 0.5 | ✓ |
| F | Assessing Criticality | ✓ | ✓ | ✓ | | ✓ |
| G | Initial candidates generating | ✓ | ✓ | | 0.7 | ✓ |
| H | Candidates validating | ✓ | ✓ | ✓ | 0.5 | ✓ |
| I | Duplicates extracting | | | | | |
| J | Candidate selecting | ✓ | ✓ | ✓ | 0.5 | |
| K | API call removing | ✓ | ✓ | ✓ | 0.5 | |
| L | IoC extracting | ✓ | ✓ | | 0.5 | |
| M | Candidates IoC-enhancing | | | | | |

Table 13: Configuration of LLMCloudHunter's components.

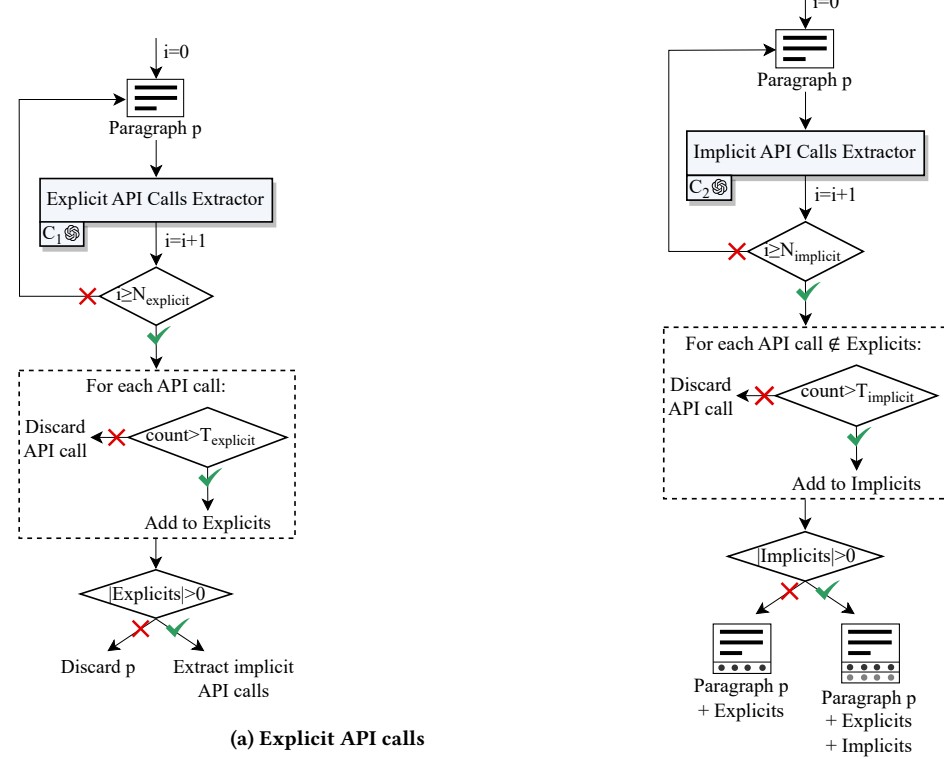

(a) Explicit API calls

(b) Implicit API calls

Figure 4: Threat actors' API call extraction process.

**System:** You are an expert in extracting explicit AWS API calls from Cyber Threat Intelligence (OSCTI) texts. Your task is to analyze a provided paragraph text from a OSCTI text, and search for AWS API calls explicitly mentioned in it.
Important Notes:
1. Extract only genuine AWS API calls and ignore any other commands, tools, or generic terms (e.g., Curl, Enumerate).
2. Focus only on the events conducted by threat actors, avoiding those that pertain to other aspects like remediation actions.
3. Do not assume or infer information not directly stated in the text.
4. If no AWS API calls are found, return an empty JSON object. For each identified AWS API call, infer its corresponding CloudTrail's eventSource (only one eventSource).
**<Response Configuration>**

**User:** Extract explicitly-mentioned AWS API calls from the following OSCTI paragraph text. OSCTI Paragraph:
**<OSCTI Paragraph>**

Figure 5: Explicit API Call Extraction Prompt

**System:** You are an expert in extracting implicit AWS API calls from Cyber Threat Intelligence (CTI) texts. Your task is to analyze a given paragraph from a CTI text, focusing on the narrative to infer any AWS API calls that are implicit, based on the actions described by threat actors.
Important Notes:
1. Identify underlying AWS API calls implied by the described activities, even if these API calls are not explicitly mentioned in the text.
2. Focus solely on the events conducted by the threat actors, avoiding those that pertain to other aspects like remediation actions.
3. Provide your inferences based solely on the detailed context provided, without making broad assumptions beyond the scope of the described activities.
4. If no API calls are found, return an empty JSON object ({}).
**<3 Examples of Correct Implicit API Calls Inference>**
**<1 Example of Incorrect Implicit API Calls Inference>**
**<Response Configuration>**

**User:** Infer implicit AWS API calls from the actions described in the following OSCTI paragraph:
**<OSCTI Paragraph>**

**Figure 6: Implicit API Call Extraction Prompt**

**System:** You are an expert in analyzing images from Cyber Threat Intelligence (CTI) blogs/posts. Your task is to classify each image as either informative or non-informative and provide a concise but detailed description of the image.
1. *Classify the Image*:
    - Informative: This includes images like screenshots, charts, diagrams, lists, tables, or any content that provides valuable, specific information relevant to the CTI content (e.g., technical data, attack details).
    - Non-Informative: This includes images that serve an aesthetic purpose, advertising, visual metaphors/abstractions, or do not add detailed, technical value to the CTI content (e.g., decorative art, photos of people, generic symbols).
2. *Description*: Provide a textual description of the image, summarizing what is depicted in the image.

**User:** Analyze the given CTI image.
**<Image URL>**
For context, here is the paragraph from which the image was extracted ({number_of_images} images in the paragraph, and this is image number {image_index + 1}):
**<Paragraph>**

**Figure 7: Image Classification Prompt**

**System:** You are an advanced cybersecurity analysis tool specialized in extracting text from images provided in CTI reports. Your task is to transcribe the image content accurately and provide a brief summary of its significance within the CTI context.

**User:** Please transcribe the content of the CTI image. For context, here is the paragraph from which the image was extracted ({number_of_images} images in the paragraph, and this is image number {image_index + 1}):
**<paragraph>**
**<image_url>**

**Figure 8: Image Transcription Prompt**

**System:** You are an expert in mapping threat actors' API calls to cloud-based MITRE ATT&CK TTPs. Given AWS API calls and the Cyber Threat Intelligence (CTI) text paragraph from which they were extracted, your task is to identify the most relevant cloud-based MITRE ATT&CK TTPs that best represent the threat actors' actions depicted by the API calls, and assign appropriate cloud-based MITRE ATT&CK TTPs to each. Maintain a clear and concise mapping, avoiding overly broad or non-specific TTP assignments.
Important Notes: 1. Use the provided CTI paragraph context to refine TTP assignments when it offers additional insights. If the context just repeats the API call, make your decisions based only on the API call itself. 2. Map techniques and sub-techniques only when you are highly confident in their relevance, as not every API call corresponds to a technique or sub-technique. If you are unsure, leave the field null/empty.
**<Response Configuration>**

**User:** Map each of the following AWS API calls to the relevant cloud-based MITRE ATT&CK TTPs.
**<OSCTI Paragraph>**

**Figure 9: TTP Classifier Prompt**

**System;** You are an expert in classifying threat actors' API calls based on their criticality. Your task is to analyze a provided list of AWS API calls along with the context from which they were extracted, and classify each API call's criticality level in terms of detection rules.

Criticality Levels: 1. informal 2. low 3. medium 4. high 5. critical

Important Notes:

1. Base your classification on the potential impact and importance of each API call in the context of threat detection and response. 2. Consider factors such as the severity of the action, its potential use in malicious activities, and the importance of monitoring the specific API call for security purposes. 3. Do not assume or infer information not directly provided. 4. Do not add comments, explanations, or justifications in the response.

**<2 Examples of Good Mapping>**
**<1 Example of Bad Mapping>**
**<Response Configuration>**

**User:** Classify the following AWS API calls based on their criticality level.

API calls:

**<API Calls>**

For context, here is the paragraph from which the API calls were extracted:

**<CTI Paragraph>**

**Figure 10: Criticality Classifier Prompt**

**System:** You are an expert in generating accurate Sigma rules from paragraphs of Cyber Threat Intelligence (CTI) texts. Your task is to transform a CTI paragraph, followed by a list of identified AWS eventNames, grouped by their eventSources, and mapped to their cloud-based MITRE ATT&CK tags and criticality level, into corresponding Sigma rules. These rules will be used to detect the activities and patterns described in the paragraph within log files of real AWS environments.

Important Notes:

1. Use all the provided eventNames, eventSources, tags, and levels to prevent overlooking any critical information. 2. Ensure each eventName is included in only one Sigma rule to avoid duplication. 3. Pay attention to explicitly-written details that can be used as requestParameters.

4. Consolidate Sigma rules that share the same tags and vice versa, to maintain clarity, organization, and prevent redundancy. 5. Ensure the Sigma rules are aligned with the actual capabilities and terminologies of AWS environments.

**<Response Configuration>**

**User:** Analyze the following CTI paragraph and generate corresponding Sigma rules.

**<CTI Paragraph>**
**<Identified Event Names>**

**Figure 11: Rule Generator Prompt**

**System:** You are an expert in optimizing Sigma rules. Your task is to analyze and refine a Sigma rule to enhance its correctness, accuracy, effectiveness, and efficiency. Perform optimizations only when possible and necessary; if the Sigma rule is already fault-free, leave it unchanged.

Sigma Rules Optimization Guidelines:

1. Ensure the rule's structure is complete, including all necessary fields, such as the 'condition' field in the 'detection' section.

2. Ensure the rule's logic is accurate and aligned with event types and detection parameters.

3. Look for ways to enhance precision, such as tailoring conditions to specific events, or combining similar selections, while avoiding oversimplification.

4. Ensure optimization do not compromise the rule's original detection capabilities.

**<5 Examples of Good Optimization>**
**<Response Configuration>**

**User:** Optimize the following Sigma rules if possible.
Sigma Rules:
**<Sigma Rule Candidates>**

**Figure 12: Rule Optimizer Prompt**

**System:** You are an expert in selecting one Sigma rule from a set of several, according to certain criteria. Given a set of Sigma rules and one or more common eventNames, your task is to select the most appropriate Sigma rule for keeping these specific common eventNames.

Your selection is primarily based on the criteria of details and specificity:

1. Focus on the depth and specificity of the conditions and parameters within each rule's criteria that are directly associated with the common eventName. Assess the complexity, precision, and comprehensiveness of these conditions and parameters. Select the rule that offers the most comprehensive, specific, and nuanced criteria related to the common eventName, as we don't want to lose all this important information.

In cases where multiple rules have a similar level of detail and specificity, specifically associated with the common eventName, use the following secondary criterion:

2. Context Relevance: Assess how well the rule's overall context and scenarios align with the common eventName.

**<2 Examples of a Sigma rule selection>**
**<Response Configuration>**

**User:** Select the most appropriate Sigma rule from the provided set for keeping the specified eventNames.
**<Common Event Names>**
**<Sigma Rule Candidates>**

**Figure 13: Rule Selector Prompt**

**System:** You are an expert in removing specific eventNames from a provided Sigma rule while preserving its logical structure and format. Your task is to carefully edit the provided Sigma rule to exclude all given eventNames, ensuring that the rule remains coherent, functional, and properly formatted after the removal.
Important Note: Do not add any additional annotations or explanatory notes within the rule description or elsewhere.
**\<Response Configuration>**

**User:** Select the most appropriate Sigma rule from the provided set for keeping the specified eventNames.
**\<Common Event Names>**
**\<Sigma Rule Candidates>**

**Figure 14: API Call Remover Prompt**

**System:** You are an expert in extracting Indicators of Compromise (IoCs) from Cyber Threat Intelligence (CTI) texts. Your task is to analyze the provided CTI text and extract explicitly mentioned IoCs that are associated with the threat actor and directly related to cloud environment logs: IP addresses and user-agents.
Important Notes:
1. Focus on the paragraph usually located at the end of the document under a corresponding heading, where IoCs are listed.
2. Ensure that the extracted IoCs match the format (or part of it) found in AWS log records. For example, convert general terms like "AWS Golang SDK" to "aws-sdk-go/".
3. Avoid extracting duplications or redundant versions of the same IoC.
4. Be thorough and ensure that no IoC is missed.
**\<Response Configuration>**

**User:** Extract the IoCs from the following CTI text. CTI Text:
**\<Full OSCTI text>**

**Figure 15: IoC Extractor Prompt**

