# OpenReview forum: "LLMCloudHunter: Harnessing LLMs for Automated Extraction of Detection Rules from Cloud-Based CTI"
_ACM.org/TheWebConf/2025/Conference — WWW 2025 Poster_

### Official Review · Reviewer_ccnP · 2024-11-21

**Novelty:** 3
**Technical Quality:** 2

**Review:**

This paper introduces LLMCloudHunter, an end-to-end framework leveraging large language models (LLMs) to analyze textual and visual Open-Source Cyber Threat Intelligence (OSCTI). The framework allows for updates and improvements without requiring fine-tuning. For experiments, the authors evaluated the quality of rules generated by LLMCloudHunter using 20 annotated cloud threat reports.

**Questions:**

1. The paper primarily represents an engineering effort. Compared to prior work, its main contribution lies in incorporating image processing into the OSCTI preprocessing stage. Technically, it relies on prompt design and LLM invocation, which limits the originality.
2. Although the related work section provides an overview, the experimental section does not include comparisons with previous methods.
3. Efficiency and Feasibility Analysis: Given that the proposed framework heavily relies on LLM invocation, an analysis of its efficiency and feasibility should be included to demonstrate its practical applicability.
4. Attention should be given to typos, such as the subheading "Image Analysys" in Section 3.1.

**Reviewer Confidence:**

4: The reviewer is certain that the evaluation is correct and very familiar with the relevant literature

**Scope:**

2: The connection to the Web is incidental, e.g., use of Web data or API

---

### Official Review · Reviewer_VcVv · 2024-11-23

**Novelty:** 6
**Technical Quality:** 4

**Review:**

__Pros:__
1. This type of work has the potential to make an immediate and significant impact on industry systems. Typically, these systems rely on detection updates derived from Open Source Cyber Threat Intelligence (OSCTI) published by various organizations. However, the current approach to utilizing OSCTI comes with a significant limitation: it is neither scalable nor easily automated, which restricts its effectiveness in large-scale threat detection efforts.  To overcome this challenge, LLMCloudHunter provides an innovative solution. It processes unstructured OSCTI content, such as blog posts, and automatically generates a set of Sigma rules. These rules can then be directly applied to enhance threat detection capabilities, making the process more efficient, scalable, and automated.
2. The paper is well-written and presented in a clear, concise manner, making it easy to follow and understand.
3. The paper demonstrates a good use case for LLMs, showcasing their potential to create a highly scalable approach for extracting signatures from OSCTI reports. This method could significantly streamline and automate the process, enhancing its efficiency and applicability across various use cases.

__Cons:__
1. The evaluation of LLMCloudHunter was conducted on a very limited dataset, consisting of only 20 samples. This is surprising, given that hundreds of OSCTI posts are published annually (for example, see [1]). The small sample size raises questions about the robustness of the evaluation. Specifically, it leads to concerns about whether the samples were carefully selected—or potentially cherry-picked—to artificially inflate the system’s accuracy. While there may be no intention of bias in the sample selection, this assumption needs to be verified.  A more extensive and diverse dataset should be used in future evaluations to ensure the system's performance is representative of many different scenarios. This is especially important in the context of OSCTI, since so many different formats will be received as input to LLMCloudHunter. Expanding the dataset would not only improve the confidence in the results but also better demonstrate the scalability and effectiveness of LLMCloudHunter.

2. There are several existing systems that attempt to extract OSCTI content, as described in the related work. However, the authors do not compare LLMCloudHunter to any of these systems, which makes it difficult to fully measure how much of an improvement their approach would be compared to prior work. It would be nice if the authors could provide some additional experiments that could demonstrate the superiority of their system over the exsiting work. However, based on the description of the related work, it is unclear how straightforward this is, so this concern is less concerning than the dataset described in issue 1. However, if they could do this it would greatly improve the contribution of the work.

[1] https://github.com/security-kg/oscti-data/tree/main

**Questions:**

1. How straightforward would it be to incorporate additional samples (e.g., around 100) into the evaluation?__

2. Would it be possible to release the reports used in the evaluation on an anonymous GitHub repository? This would allow others to verify the diversity of the dataset and ensure the evaluation was conducted on a representative sample.

3. Why does the paper focus exclusively on cloud-based systems? How challenging would it be to adapt LLMCloudHunter to write rules for more generalized systems, such as firewalls? The emphasis on cloud systems seems somewhat limiting, and the intent behind this narrow focus isn't entirely clear to me. Broadening the scope to include non-cloud environments could make the tool more versatile and impactful, potentially benefiting a wider range of security applications.

4. How difficult would it be to adapt LLMCloudHunter to support signature-based systems beyond Sigma? Many organizations rely on custom in-house signature detection systems, so adding flexibility to accommodate different formats could significantly enhance the tool's utility and adoption. What would be involved in updating the system to generate rules compatible with a variety of signature frameworks?

**Reviewer Confidence:**

3: The reviewer is confident but not certain that the evaluation is correct

**Scope:**

4: The work is relevant to the Web and to the track, and is of broad interest to the community

---

### Official Review · Reviewer_EPx6 · 2024-11-27

**Novelty:** 4
**Technical Quality:** 3

**Review:**

This paper proposes LLMCloudHunter, an end-to-end framework designed to analyze OSCTI (Open-Source Cyber Threat Intelligence) using pre-trained LLM models. LLMCloudHunter can process both text and image formats of OSCTI, automatically extracting key entities and relationships, and generating detection rules in Sigma format. The key components of the framework include API call extractors, IoC extractors, TTP classifiers, rule optimizers, etc. These components are optimized for efficiency and accuracy through parallelization and few-shot learning techniques. The output of LLMCloudHunter can be seamlessly integrated into existing SIEM systems, demonstrating its strong flexibility and scalability. Experimental results show that LLMCloudHunter performs excellently in entity and relationship extraction accuracy and Sigma rule generation, particularly achieving high precision and recall in API call and IoC extraction tasks.

Strengths:
1. **Innovation**: The framework combines the powerful capabilities of pre-trained LLMs and innovatively applies them to the OSCTI analysis field, enhancing threat detection, especially through image processing and text extraction techniques.
2. **Efficiency and Scalability**: Through parallel processing and few-shot learning techniques, LLMCloudHunter significantly increases processing speed and efficiency, making it highly suitable for large-scale datasets. Additionally, the framework can flexibly adapt to different organizational environments and OSCTI formats, offering good scalability.
3. **Complete Framework Design**: LLMCloudHunter demonstrates a comprehensive end-to-end process, from extracting image and text data from OSCTI, identifying entities and relationships, to generating Sigma rules, showcasing its potential for practical application.
4. **Experimental Validation**: Extensive experiments validate the framework’s effectiveness across multiple OSCTI tasks, providing detailed evaluation results, especially excelling in API call extraction and MITRE ATT&CK label extraction.
5. **Practical Application**: Since the output of LLMCloudHunter is directly compatible with SIEM systems, it holds significant practical value, helping security teams to detect and respond to threats more efficiently.

Weaknesses:
1. **Dependency on Image Processing**: Although image classifiers and image transcribers play a key role in information extraction, their reliance on image content means the framework's performance may be limited when valid image information is unavailable. Therefore, the practical application of the framework may require ensuring the quality of image data.
2. **Lack of Model Diversity**: The paper only uses the open-source GPT model as the backbone, without exploring other LLM models. Implementing multiple backbone models would enhance the robustness of LLMCloudHunter.
3. **Lack of Diversity Testing**: The experimental data mainly focuses on testing with 20 OSCTI samples. Future work could test the framework’s generalization ability using a broader dataset, particularly when confronted with more complex or unknown attack patterns. The sample dataset size and statistics presented in the paper should be compared with previous datasets to demonstrate the applicability of the dataset used in this study.

**Questions:**

1. The dataset is relatively small—can this demonstrate the effectiveness of LLMCloudHunter?
2. Traditional security methods often struggle with the dynamism and distributed nature of cloud environments. Why do LLM-based methods offer a better approach to tackling the challenges of the cloud environment? Please explain to help better understand the motivation behind this paper.

**Reviewer Confidence:**

2: The reviewer is willing to defend the evaluation, but it is likely that the reviewer did not understand parts of the paper

**Scope:**

3: The work is somewhat relevant to the Web and to the track, and is of narrow interest to a sub-community

---

### Official Review · Reviewer_FXDS · 2024-12-02

**Novelty:** 6
**Technical Quality:** 6

**Review:**

Strengths
1. Comprehensive Framework: Integrates textual and visual CTI for actionable Sigma rule generation, addressing gaps in cloud-based threat intelligence analysis.
2. Accessibility: Outputs in Sigma format ensure broad compatibility with SIEM systems.
3. Clarity: Clear description of pipeline phases and innovative use of techniques like few-shot learning and parallel processing.

Weaknesses
1. Limited Real-World Testing: The framework's effectiveness in dynamic and adversarial cloud environments remains untested.
2. Computational Overhead: The paper does not sufficiently discuss the cost and scalability of LLM usage in resource-constrained scenarios.
3. Criticality Assignment: The criticality classifier's accuracy (average 88%) suggests room for improvement in aligning with expert assessments.

Suggestions for Improvement
1. Test the framework in real-world deployment scenarios, particularly with adversarial inputs or noisy data.
2. Provide a detailed analysis of the computational cost and potential optimizations for large-scale or real-time applications.
3. Enhance the criticality classifier's methodology to align more closely with expert judgments.

**Questions:**

Questions for the Authors
1. How does the framework perform in real-world cloud environments, particularly with adversarially crafted CTI?
2. What are the scalability implications of using LLMs for large-scale or real-time CTI analysis?
3. Can the framework be adapted for other domains or types of CTI, such as IoT or industrial control systems?

**Reviewer Confidence:**

4: The reviewer is certain that the evaluation is correct and very familiar with the relevant literature

**Scope:**

4: The work is relevant to the Web and to the track, and is of broad interest to the community

---

### Official Review · Reviewer_vsLM · 2024-12-02

**Novelty:** 5
**Technical Quality:** 5

**Review:**

## Paper Summary

This paper introduces LLMCloudHunter, a novel framework that leverages LLMs to automate the extraction of detection rule candidates from unstructured open-source cyber threat intelligence (OSCTI) in cloud environments. By integrating textual and visual data, LLMCloudHunter processes unstructured OSCTI into structured, actionable detection rules, achieving high precision and recall in identifying API calls, IoCs, and related relationships.

## Detailed comments

### Pros

- **Clarity**: The work is readable and well presented
- **Significance**: The work fills a critical research and operational gap for cloud environment threat hunting.
- **Novelty:** Some novel techniques such as visual information.

### Cons

**Heavy Reliance on LLM Performance**:

- The results depend highly on model capabilities, and commercial LLMs may be less scalable.
- Can other LLMs work well similarly? It may be beneficial to test on an additional open-source LLM.

**Evaluation can be improved**:

- The work focuses exclusively on cloud environments without validating on hybrid or on-premises setups. Can LLMCloudHunter also apply to the other settings?
- Can it be compared with other related work not focusing on cloud environments? Without some comparison, it may be hard to draw that it is superior to other methods under cloud or on-premise setting.

**Rooms of Improvement for TTP Extraction**:

- Low precision in mapping MITRE ATT&CK TTPs indicates room for improvement in contextual alignment.

**Questions:**

1. Line 678 - 696: For evaluation like "Descriptive metadata alignment", it is mentioned that "(they) were calculated by our research team." Does this mean the these are human evaluated?
2. Have you tested the framework in real-world threat hunting scenarios or received feedback from practitioners?
3. Dataset annotation: How was the ground truth dataset annotated, and what measures were taken to ensure consistency across annotators?

**Reviewer Confidence:**

2: The reviewer is willing to defend the evaluation, but it is likely that the reviewer did not understand parts of the paper

**Scope:**

3: The work is somewhat relevant to the Web and to the track, and is of narrow interest to a sub-community